# Unsupervised approach to decomposing neural tuning variability

Rong J. B. Zhu [1,2] & Xue-Xin Wei [3,4,5,6]

Neural representation is often described by the tuning curves of individual neurons with respect to certain stimulus variables. Despite this tradition, it has become increasingly clear that neural tuning can vary substantially in accordance with a collection of internal and external factors. A challenge we are facing is the lack of appropriate methods to accurately capture the moment-to-moment tuning variability directly from the noisy neural responses. Here we introduce an unsupervised statistical approach, Poisson functional principal component analysis (Pf-PCA), which identifies different sources of systematic tuning fluctuations, moreover encompassing several current models (e.-g.,multiplicative gain models) as special cases. Applying this method to neural data recorded from macaque primary visual cortex– a paradigmatic case for which the tuning curve approach has been scientifically essential– we discovered a simple relationship governing the variability of orientation tuning, which unifies different types of gain changes proposed previously. By decomposing the neural tuning variability into interpretable components, our method enables discovery of unexpected structure of the neural code, capturing the influence of the external stimulus drive and internal states simultaneously.

A central goal in neuroscience is to determine how neural responses depend on external stimulus variables and the internal states of the brain. The dependence of individual neuron's firing rate on a stimulus variable is often described by the turning curve, i.e., the average firing rate of a neuron as a function of the stimulus[1–4]. Because tuning curves are the consequences of various internal computations in neural circuits, it is likely and indeed empirically the case that it could be modulated by factors other than the stimulus variable selected a priori[5–21]. Indeed, variability of neural tuning has now been widely reported in neural systems, and furthermore been proposed to exhibit various forms, including multiplicative gain[6,18,22–27], additive modulation[5,28–32], shift of tuning peaks[33–35], and tuning width changes[36]. These observations reflect the role of various factors, whether related to stimuli (e.g., stimulus contrast[37], stimulus history[34,38]), behavior (e.g., movement[16]), or latent brain states[25,28].

Tuning variability has been widely implicated both functionally, i.e., information encoding and the behavioral performance[15,19,28,30,39,40], and mechanistically, i.e., how tuning variability is generated[41–43]. Prior studies have attempted to quantitatively model the variability of tuning in the sensory cortex, in particular orientation tuning in the primary visual cortex (V1), which is widely considered a paradigmatic case for studying neural code. Decades of studies in V1 have shed general insights regarding how neurons in the cortex encode external sensory variables. Perhaps surprisingly, studies of V1 tuning variability have yielded results that are seemingly at odds so far. One line of work[25,26] has proposed a simple multiplicative gain model to account for the tuning variability. Multiplicative gain has been postulated to have a vital role in encoding contrast[44], encoding uncertainty[27], facilitating downstream readout[45], implementing attention[6,7], as well as the transformation of coordinate systems (e.g., retina- to body-centered)

[1]Institute of Science and Technology for Brain-Inspired Intelligence, Fudan University, Shanghai, China. [2]MOE Key Laboratory of Computational Neuroscience and Brain-Inspired Intelligence, and MOE Frontiers Center for Brain Science, Shanghai, China. [3]Department of Neuroscience, The University of Texas at Austin, Austin, USA. [4]Department of Psychology, The University of Texas at Austin, Austin, USA. [5]Center for Perceptual Systems, The University of Texas at Austin, Austin, USA. [6]Center for Theoretical and Computational Neuroscience, The University of Texas at Austin, Austin, USA. ✉e-mail: rongzhu@fudan.edu.cn; weixx@utexas.edu

in parietal cortex[46,47]. Mechanistic models suggest that multiplicative gain could result from threshold-linear neurons operating in the presence of intrinsic intracellular noise[41–43]. In contrast to the above work, other studies have suggested additive interactions[5,18,48] or both additive modulation and multiplicative gain[28,30,31] in V1.

Crucially, the analysis methods in the majority of this prior work presumed relatively restrictive structure for tuning variability (e.g., refs. [18,25,26,28,30,32]), leaving open the question of whether other forms of fluctuations might in fact account for the data better. Furthermore, existing analyses generally relied on trial-averaging and comparison across conditions[6,30,38], thus failing to capture the moment-to-moment variability in tuning. Addressing these open issues requires approaches that can infer the structure of tuning fluctuations directly on single-trial data—and ideally on the raw spike train itself—while also avoiding restrictive assumptions.

Here we introduce an unsupervised statistical technique, Poisson functional PCA (Pf-PCA), to identify the structure of of latent tuning fluctuations directly from neural spiking data. Importantly, we apply this method to address tuning variability in a classic neural system that has long been characterized via tuning, namely neurons in V1. Because Pf-PCA yields a generative model of the moment-to-moment tuning variability, where a moment is defined by a block consisting of responses for all stimuli, it could be used to analyze the information encoding through the calculation of information-theoretical measures such as Fisher information. It could also be used to analyze the

geometrical structure of the neural manifold. Performing these analyses, we find that Pf-PCA reveals several insights. The proposed analysis framework is broadly applicable to other low-dimensional tuning modalities.

## Results

Previous studies have suggested that the tuning fluctuations may be heterogeneous[30,38]. This heterogeneity motivated us to develop a flexible, unsupervised analysis framework to understand the tuning variability. It may be used to analyze the tuning fluctuations of any one-dimensional variable with smooth tuning properties. We will first develop and validate our method, and then apply it to analyze Macaque V1 data. We will show that our method helps reveal insights into the structure of the neural code for visual orientation, the information content, and the geometry of the representation.

### The Poisson functional PCA framework

Figure 1a illustrates the basic modeling framework of Pf-PCA (see "Methods" for details). The model assumes that the logarithm of the tuning curves (of an arbitrary stimulus variable) is determined by a smooth mean component and smooth functional principal components (fPCs) weighted by the amount of latent fluctuations. Note that each fPC is a function that is tuned to the stimulus variable. The fPCs and their weights (i.e., scores) together capture the fluctuations of the tuning curves. Quantitatively, the tuning curve $\boldsymbol{\mu}_t$ for the $t$-th moment

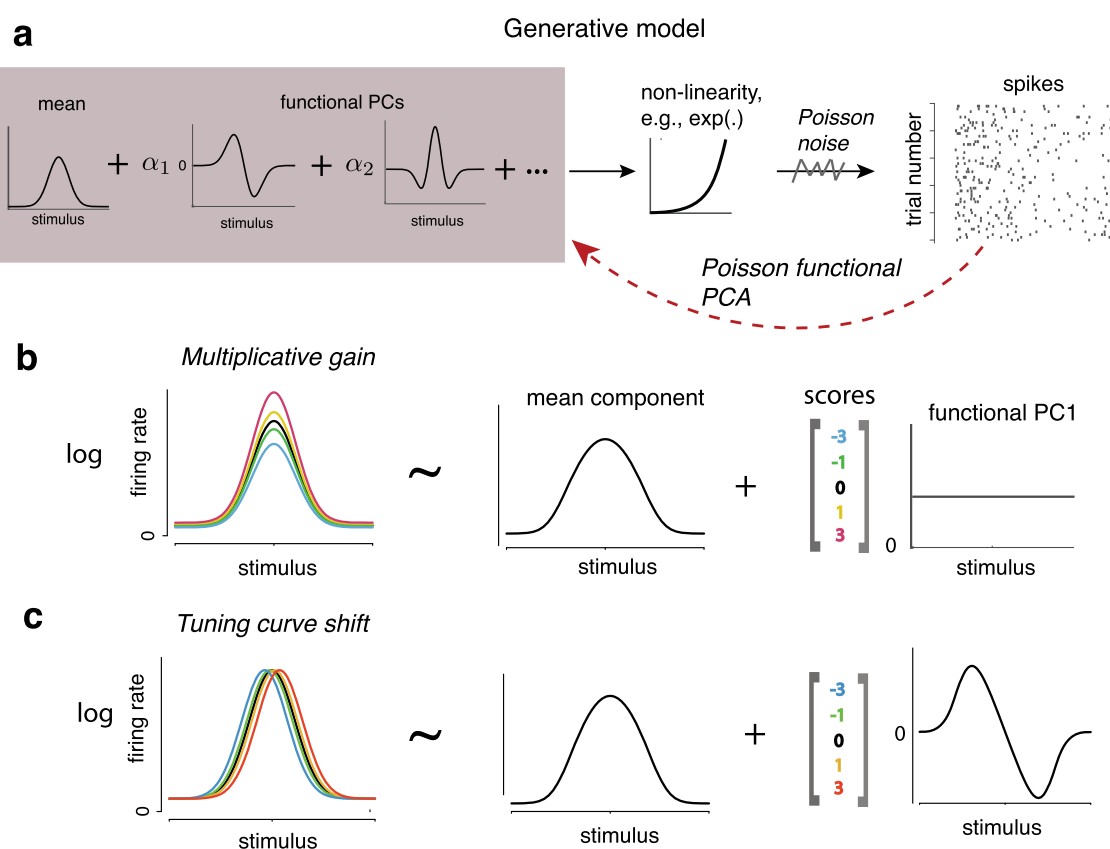

**Fig. 1 | The Poisson functional PCA (Pf-PCA) framework. a** Illustration of the Pf-PCA framework. Tuning functions (i.e., the logarithm of the tuning curves) are modeled as a sum of the mean and functional principal components (fPCs) weighted by the amount of latent fluctuations $\alpha$ (i.e., scores). Tuning functions then pass through a static non-linearity to obtain the tuning curves. The non-linearity is assumed to be an exponential function in this study. Finally, standard Poisson spiking noise is assumed for the generation of the observed spike trains. The Pf-PCA algorithm performs inference on the observed spike counts. It extracts the mean component, the fPCs, and the moment-to-moment scores of each fPC.

**b, c** Schematics illustrating this framework using two special cases: multiplicative gain and tuning curve shift. **b** For multiplicative gain, the logarithm of individual tuning curves (illustrated using Gaussian functions plus a baseline) can be decomposed into a mean component and a flat function principal component. Different tuning curves correspond to different scores. **c** For the case of tuning curve shift, the logarithm of the individual tuning curves can be approximated by the summation of the mean component, the first functional PC that has an anti-symmetric shape, and a small residual.

can be describe as

$$\log(\boldsymbol{\mu}_t) = \boldsymbol{f} + \sum_k \alpha_{k,t} \phi_k + \boldsymbol{\epsilon_t}. \tag{1}$$

Here $\boldsymbol{f}$ is the mean component, $\phi_k$ is the $k$-th fPC, $\alpha_{k,t}$ denotes the amount of fluctuation (i.e., score) for the $k$-th component during the $t$-th moment, and is assumed to follow a zero-mean Gaussian distribution. The last term $\boldsymbol{\epsilon_t}$ is assumed to be zero-mean Gaussian noise that captures the residual unstructured fluctuations. The spike train at every moment is assumed to be generated from a Poisson process with the firing rate specified by Eqn. (1). Note that with only the first term $\boldsymbol{f}$, this model is equivalent to the standard tuning curve model of spike counts. The remaining terms captures the additional variance from the contribution of moment-to-moment fluctuations, making the model naturally capture the over-dispersion of spike counts[25,32].

Our algorithm takes spike count data as the input and infers the mean, fPCs, the variance of each component, as well as the weight for each fPC for each moment. Critically, the shape of fPCs, which specifies the particular form of the fluctuations, is directly inferred from the data. When studying the neural response to continuous stimulus variables, it is natural to assume the mean component and fPCs of individual neurons as some smooth functions of stimulus. Importantly, our method merely assumes that the mean component and fPCs are smooth, without imposing restrictive assumptions on their shapes. Our method provides a way to parse the variability of tuning into a set of fPCs from the spike counts. Now consider a couple of special cases. With the additional assumptions that there is only one fPC and that it is constant over the stimulus dimension (Fig. 1b), our model becomes essentially the multiplicative gain model[25]. When tuning curves exhibit systematic lateral shifts, our model could capture it with an fPC that is proportional to the derivative of the mean component (Fig. 1c). It is worth emphasizing that our method is general and capable to capture other cases that may be potentially more complicated. It may be used to analyze the tuning fluctuations of any one-dimensional stimulus variables with smooth tuning properties.

Our method is developed by adapting a technique, i.e., functional PCA[49–53], to deal with Poisson spiking noise. As the firing rate is not observed, it makes the inference procedure more challenging. We resolve this problem by developing a procedure based on an Expectation-Maximization algorithm. Details of the inference procedure are described in the Methods section. In broad strokes, the algorithm treats the unobservable firing rates of stimulus $s$ $\mu_t(s)$ as "observations" generated from the model in Eqn. (1). To maximize the likelihood, the algorithm iterates between estimating the mean and covariance matrix parameters and calculating their posteriors based on the spike counts given their current estimates via the Monte Carlo method. This step gives an estimate of the firing rates. In the next step, we apply the functional PCA technique to the estimated firing rates for estimating the components and the moment-to-moment fluctuations.

### Validation of the method

We validated our method systematically using simulated data. Inspired by previous experimental observations on tuning variability[22,23,25,28,30,33–36], we first examined whether our method is able to recover fPCs that correspond to multiplicative gain, additive change, tuning shift, or sharpening. Specifically, we generated synthetic data, which exhibit different types of tuning fluctuations by reverse-engineering the appropriate fPC, and tested Pf-PCA and alternative methods with these data, where the ground-truth were known.

Figure 2a shows results based on the analysis of the simulated datasets using our method and alternative methods (see "Methods" for details). We found that Pf-PCA could accurately recover the form of the fluctuations in all four cases. Furthermore, it approximately recovers

the proportions of variance explained by the structured fluctuation (Fig. 2a), as well as the magnitude of the latent fluctuation on a moment-by-moment basis.

How does our method compare to simpler methods? Applying conventional PCA to the synthetic data, we found that it often misidentified the form of the fluctuation, and that it could not reliably estimate the magnitude of the latent fluctuations (Fig. 2b). We also applied a variant of our method, referred to as $\mu$-PCA, by removing the smoothness constraint in our full algorithm (see "Methods" for details). This algorithm is similar to the Poisson PCA[54] (a discussion of the technical differences between $\mu$-PCA and Poisson-PCA can be found in "Methods"). The $\mu$-PCA generally performs better than regular PCA, but is still considerably worse than the full method Pf-PCA.

We further validated our method when multiple types of fluctuations co-exist, e.g., a combination of multiplicative gain and tuning shift. We found that Pf-PCA could recover both components reliably (see Supplementary Fig. 3), and that it drastically outperforms regular PCA and $\mu$-PCA (see Supplementary Fig. 4). In addition, we validated our method in the case of monotonic tuning curves, e.g., the sigmoidal tuning curves, and found similar results (see Supplementary Fig. 5). Taken together, these results on synthetic data suggest that our method could robustly recover the structure and magnitude of the tuning fluctuations using an experimentally realistic amount of data.

### Pf-PCA reveals power-law modulation of neural tuning

We next show that our method can be used to reveal scientific insights into neural codes. We will focus on the variability of orientation tuning in macaque V1, which has been a question of substantial interest in the past decades and may have general implications regarding the principle of neural coding in the cortex. Previous studies have mainly focused on "gain variability", which assumes constant additive modulations or multiplicative gain that scales the whole tuning curve. The nature of this tuning variability has been heavily debated to date. Our unsupervised approach enables us to generalize the notion of "gain variability" to general "tuning variability", resulting in a more accurate understanding of the structure of the neural response.

We analyzed seven previously published datasets, each with dozens of neurons simultaneously recorded from macaque V1[30,55] (total number of neurons = 402). During these experiments[30,55], drift gratings with different directions were presented, each for 1 or 1.28 seconds. A block-randomized design was used in these experiments, with each block sampled a pre-determined set of stimulus directions once. See "Methods" for details. To build some intuitions on orientation tuning variability, we first split the total blocks into two halves according to the number of spikes for individual neurons, and calculated the tuning curves for the high and low conditions[30]. Figure 3a shows six representative example neurons. Visual inspections suggest that tuning variability is heterogeneous across neurons, exhibiting features consistent with an additive modulation or multiplicative gain or both, though other times neither.

We applied Pf-PCA to analyze the tuning fluctuations for stimulus orientation. We treated each block of stimuli as one moment, assuming that the tuning curve is stable within each block. Thus the tuning fluctuations studied here is at the timescale of ~10 s. The Pf-PCA model achieves a better fit compared to the Modulated Poisson model that assumes a multiplicative gain[25], assessed through cross-validated prediction error and cross-validated likelihood (see Supplementary Note 1, Supplementary Fig. 1). When applying Pf-PCA, we assumed three fPCs, which are sufficient to capture most of the tuning variability in these data (see Supplementary Fig. 6). In fact, the first fPC alone captures 62.4% of the variance on average (Fig. 3b). Below we will focus our analysis primarily based on the first fPC.

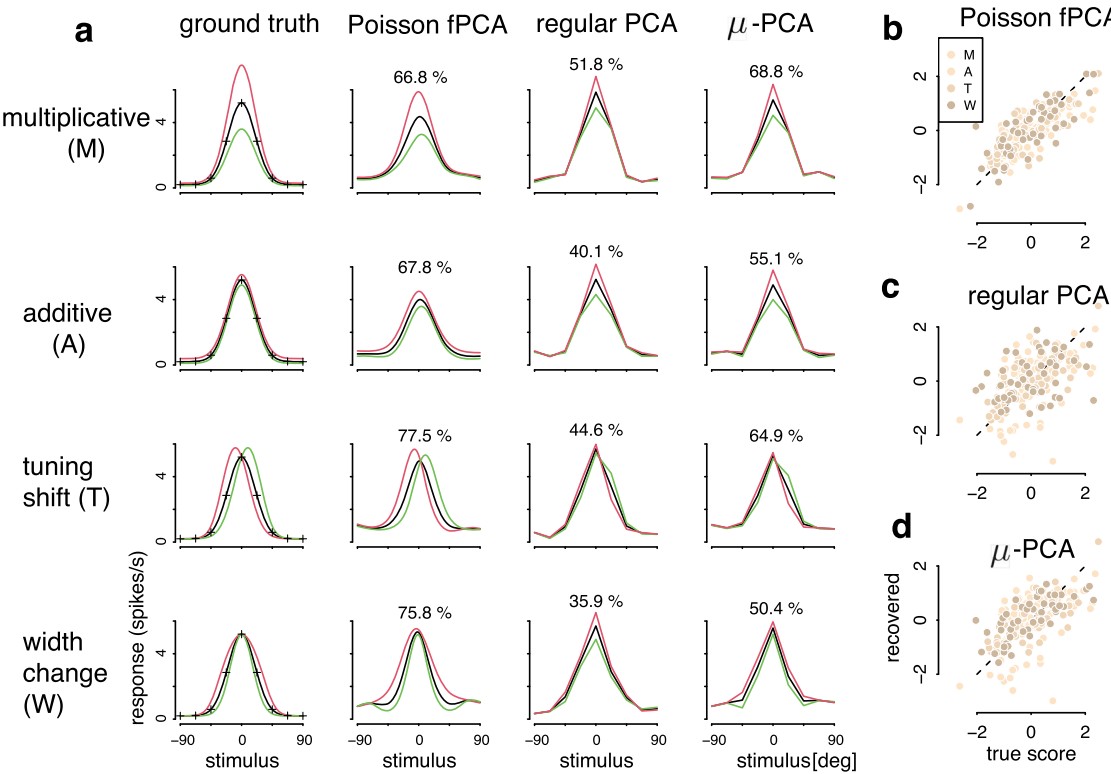

**Fig. 2 | Model validation with bell-shape tuning curves. a** Inferred fluctuations of four types of tuning fluctuations. Black curve: the tuning curve with zero fluctuation. This corresponds to the mean component exponentiated. Red/green curves: the corresponding tuning curves when setting the score to be ±1 standard deviation of the scores. Crosses in the first column: the set of discrete stimuli used in the simulation. From left to right: ground truth, Pf-PCA, regular PCA, and a reduced version of Pf-PCA termed as μ-PCA. In each case, we simulated neural responses from the corresponding ground truth model, and then inferred the form of the fluctuation and its magnitude for each moment. The percentages on the plots show the proportion of variance explained by the component. By our construction of the model, the first fPC of a "perfect" estimation procedure should explain 80% of the variance. **b–d** Recovered scores v.s true scores for each method. Light to dark bisque points denote multiplicative gain ("M"), additive modulation("A"), tuning shift ("T"), and tuning sharpening ("W"). Dashed lines indicates the diagonal. Overall, Pf-PCA substantially outperforms the alternatives in all these cases, both in terms of recovering the form of the tuning fluctuations and the magnitude at each moment. The correlations between the recovered and the ground truth are 0.788 for Pf-PCA (**b**), 0.544 for regular PCA (**c**), and 0.647 for μ-PCA (**d**).

As mentioned before, if a neuron exhibits multiplicative gain change, the first fPC should be a constant (Fig. 1b). However, we found that the first fPC for the majority of neurons is not constant (for example, see Fig. 3a). This implies that the fluctuations of the firing rate of these neurons could not be accurately described as a pure multiplicative gain, and instead the gain appears to be stimulus-dependent. Interestingly, the first fPC for most neurons is highly correlated with the mean component. This is confirmed by a simple linear regression analysis between the mean component and the first fPC (Fig. 3c). For quantification, we defined a fraction to capture the percentage of the first fPC explained by the simple linear relationship (see "Methods" for details), and found that this linear relationship explains most of the information in the first fPC (Fig. 3d). We wondered if our estimation procedure might exhibit systematic biases so that even when the ground truth model was a simple gain modulation model, the estimated first fPC might nonetheless be correlated with the mean component. We performed a control analysis and found that this is unlikely. Specifically, we simulated datasets from a multiplicative gain model with approximately matched statistics and performed Pf-PCA on the synthetic datasets (see "Methods" for details). The results showed that the slope values of the regression for the synthetic data are much closer 0 compared to what we obtained from the V1 data (Fig. 3e).

Crucially, the above observation (linear relationship between the mean tuning curve and the first fPC) has conceptually important implications for the tuning structure. In particular, the linear relationship permits the following linear approximation for the first fPC $\phi_1(s)$,

$$\phi_1(s) \approx b + wf(s). \tag{2}$$

Together with Eqn. (1) and some algebraic manipulations, we found that the tuning curve for moment $t$ can be expressed as

$$\mu_t(s) \approx \exp(b\alpha_{1,t})[\exp\{f(s)\}]^{1+w\alpha_{1,t}}.$$

Because the latent variable $\alpha_{1,t}$ appears in the exponent, it suggests that the fluctuations of the tuning curves could be in fact described as a power-law modulation, with the exponent of the power function varying from moment to moment.

## Power-law modulation accounts for both additive modulation and multiplicative gain

Previous studies have proposed two forms of gain change in V1[25,28–30], i.e., additive and multiplicative. It has been heavily debated which type of variability is more appropriate to describe the V1 activity, or whether both types of activity co-exist in V1. We hypothesize that the part of the controversy is due to the restrictive notion of gain variability in previous studies. By considering and analyzing general tuning variability as enabled by Pf-PCA, below we will demonstrate that the power-law relation unifies these different forms of gain variability.

Noticing $\mu_0(s) = \exp\{f(s)\}$ (and assuming that it is already normalized to have peak activity equal to 1 by absorbing into the intercept

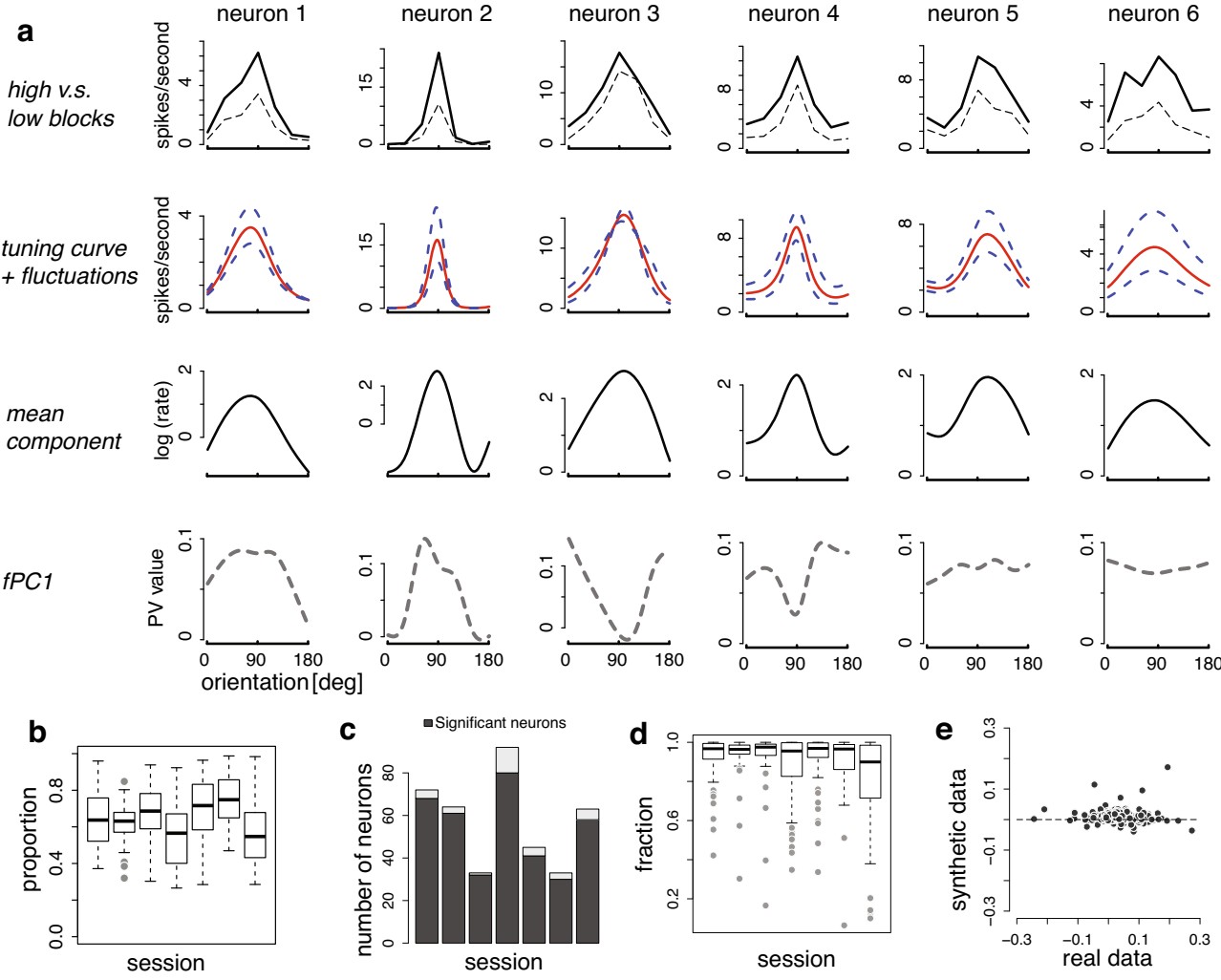

**Fig. 3 | Results of Pf-PCA on V1 data. a** Recovered mean component and first fPC for six example neurons. Neuronal tuning fluctuations exhibit a variety of structures. The top panels show the average tuning curves by splitting the experimental blocks into high and low blocks. Panels in the second row show the tuning curves and the fluctuations inferred from Pf-PCA, plotted in the firing rate space. The last two rows show the mean component and the first fPC, both plotted in the logarithm of the firing rate scale. For the last row, "PV" (y-axis) represents the value of the first fPC for each stimulus. **b** The variance explained by the first fPC. Central mark: median value. Edges of the box: 25th and 75 percentiles. Whiskers: the maxima and minima of the data within 1.5 times the interquartile range from the nearest quartile.

**c** Testing the significance of the regression analysis of the first fPC on the mean component. The majority of the neurons show a significant linear relationship between the mean component and the first fPC. **d** The "fraction" index quantifying how much information of $\phi_1(s)$ is accounted for by a linear function of $f(s)$ (see "Methods"). The observation that the "fraction" index is closed 1 for each dataset suggests that the linear function of the mean highly explains the first fPC for the majority of the V1 neurons. Box plot: similar convention to (**b**). **e** Slope values of the regression analysis obtained from real data v.s. those obtained from synthetic data. The synthetic data were generated from a multiplicative gain modulation model.

term $b$), the tuning curve on each moment can be re-expressed as

$$\mu_t(s) = \mu_0(s)^{1+w\alpha_{1,t}} \exp(b\alpha_{1,t}). \quad (3)$$

Equation (3) is a power function with the power $1 + w\alpha_{1,t}$ and the scale $b\alpha_{1,t}$, both of which are linear function of the fluctuation $\alpha_{1,t}$ for moment $t$. In this relation, for each neuron, there are two free parameters corresponding to the slope and intercept in the regression analysis respectively. Without loss of generality, we can constrain the intercept to be always non-negative. The consequence of varying each parameter on the tuning is straightforward to see. Specifically, a non-zero intercept would lead to fluctuation of the peak firing rate, while a non-zero slope would lead to systematic tuning width change due to the exponentiation (Fig. 4a). Depending on the specific combination of the slope ($w$) and intercept ($b$), the tuning fluctuation will exhibit different characteristics for individual neurons.

First, when the slope $w = 0$, the power-law modulation degenerates to a pure multiplicative gain[25]. Second, the power-law modulation could lead to approximately additive modulation with certain combinations of the slope ($w$) and intercept ($b$). For quantification, we defined a "flatness" index to characterize the change over the stimulus variable induced by the fluctuation. Informally, this index computes the ratio between the change of the firing rates between the preferred and the orthogonal orientations (see "Methods" for a formal definition). With additive modulation, the flatness index is 1, while multiplicative gain leads to a flatness index of 0. When the flatness score is negative, the resulting configurations show a sharpening of the tuning curve. Figure 4 shows the "flatness" while systematically varying the two parameters (i.e., the slope and intercept). In the appropriate parameter regimes, the power-law would manifest itself as a multiplicative or an additive change (see "Methods" for details), while parameter values in between result in tuning modulation which might be interpreted as a mix of multiplicative and additive modulations[28].

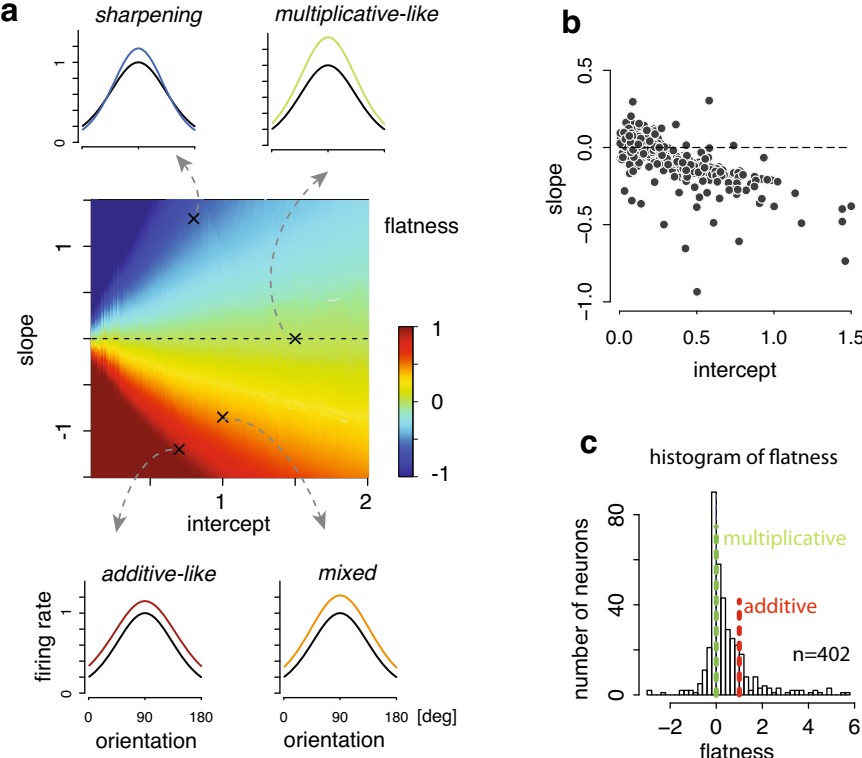

**Fig. 4 | Power-law modulation accounts for both additive and multiplicative modulations. a** The "flatness" index as a function of the slope and intercept parameters of the power-law relation. The "flatness" index quantifies the extent to which the firing rate change depends on the stimulus orientation. It manifests itself as multiplicative and additive modulations, among others, in certain parameter regimes. Multiplicative gain would lead to a zero "flatness" index, while additive change implies "flatness" equals to 1. Changing the flatness from 0 to 1 would lead to a gradual transition from multiplicative gain to additive change. This index can be smaller than 0, a case implying sharpening of the tuning curve. **b** The estimated values of slope and intercept for individual neurons. **c** The histogram for the flatness scores, indicating that tuning fluctuations lie on a continuum.

Empirically, most of the neurons lie in between multiplicative and additive changes (Fig. 4b), thus are better characterized by the proposed power-law relation than a pure multiplicative or additive modulation. The control analysis by generating simulated data using multiplicative gain showed that the recovered slope is close to zero over intercept as expected (See Supplementary Fig. 9). Note that a previous study[30] found that additive and multiplicative fluctuations were anti-correlated, which could be naturally explained by our power-law model. It is also worth mentioning that a subset of neurons exhibits a mild sharpening of the tuning curve. Together, these results provide a unified account of the fluctuations of orientation tuning in V1. Although we could not rule out the possibility of two separate mechanisms (one for multiplicative gain, and one for additive modulation), our results show that a single form of fluctuation is sufficient to capture the variability, and the tuning fluctuations of individual neurons appear to lie on a continuum.

**Population tuning fluctuations are low-dimensional**
The dimensionality of tuning fluctuations has important implications for the mechanisms and the function of the circuit. Some studies (e.g., refs. 28,29) implicitly assumed a rank-1 fluctuation that scales the gain of the population in a coherent manner, and found evidence suggesting that the total population activity is highly predictive of the moment-to-moment fluctuation of the response of individual neurons[29]. Others found the coupling strength of individual neurons to the rest of the local network to be diverse[56], implying a higher dimensionality of the tuning fluctuations and a potential role of recurrent connections in shaping network responses. Recently proposed E/I-balanced network models with spatial connectivity structure[57] predicted that the population fluctuations should be low-

dimensional. Finally, it has been proposed[27] that gain variability in V1 may serve to represent the stimulus uncertainty via sampling, a computation would generally require the gain variability to be high dimensional.

We examined the structure of the tuning fluctuation at the population level. As demonstrated earlier, for each neuron, the tuning fluctuations could be well captured by the first fPC. Exploiting this observation, we approximated tuning fluctuations of a neural population by concatenating the scores for individual neurons together (number of neurons × number of blocks, Fig. 5a). Examining the correlation of the scores, we found that while most of the neurons fluctuate coherently, a small group of neurons is anti-correlated with the rest of the neurons (Fig. 5b) in some sessions (for results for all sessions, see Supplementary Fig. 10). What is the dimensionality of latent fluctuations of the neural population? If the neurons share a coherent multiplicative change or additive change[28], the latent fluctuation should be close to one dimensional. To assess this, we performed a standard PCA analysis on the score matrix to assess the linear dimensionality. We found that, while the fluctuation shows a low-dimensional structure (Fig. 5e), the dimensionality exceeds one.

The empirically observed latent fluctuations cannot be explained by a rank-1 multiplicative or additive modulation model. To demonstrate this, we performed control analysis by generating simulated data using the rank-1 additive or multiplicative model (see "Methods" for details), and found that the resulting correlation structure of the inferred score based on the synthetic data exhibits a simpler structure (Fig. 5d, e, and Supplementary Figs. 11 and 12). The dimensionality of the scores is lower than that estimated from real data (Fig. 5f).

These results paint a more nuanced picture of the fluctuations of V1 at the neural population level. Deviating from what was suggested

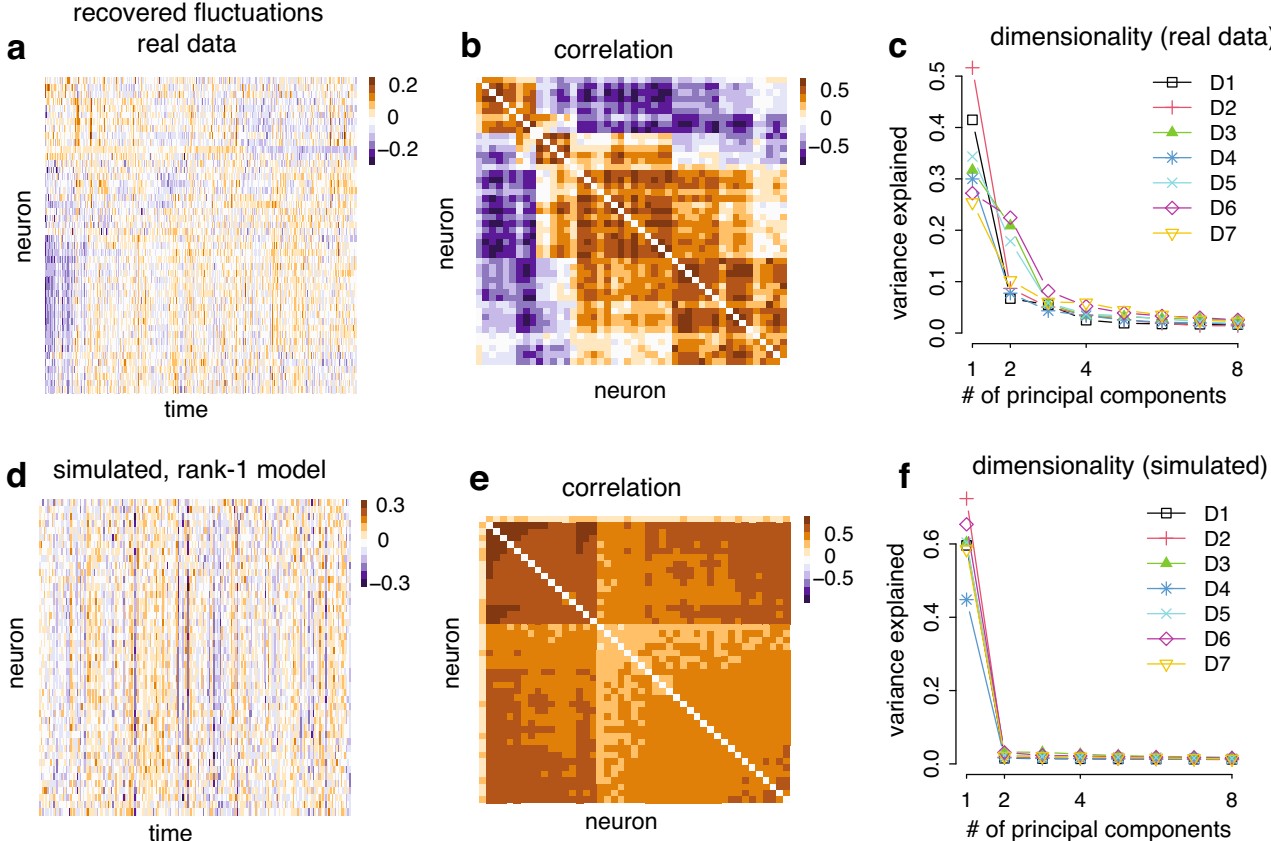

**Fig. 5 | Population structure of the tuning fluctuations: the fluctuations of V1 population as captured by the first fPC have a low dimensional structure. a** The heat map of scores of the first fPC for every neuron in one dataset (Session D5). Every data point represents the score of one neuron during one block of stimulus presentation. **b** The correlation of the score matrix above. The neurons are sorted according to a hierarchical clustering algorithm in (**a**) and (**b**). **c** The dimensionality of the fluctuations as captured by the first fPC. We run a standard PCA on the score matrix. The first two principal components capture more than 60% of the variance. **d**, **e** Similar to (**a**, **b**), but based on the simulated dataset assuming rank-1 multiplicative gain fluctuations. **f** Similar to (**c**), but based on the simulated dataset assuming rank-1 multiplicative gain fluctuations. In this case, the first principle component consistently dominates other components in the recovered score matrix. These results suggest that the fluctuations in the neural population are low-dimensional, but not one-dimensional.

previously[28], the fluctuations of V1 neurons are not completely coherent in the anesthetized state, with subset of neurons could exhibit fluctuation at the opposite direction compared to the majority, nor can it be characterized by a rank-1 additive modulation or multiplicative gain fluctuations. These results will help further constrain and refine the network mechanisms giving rise to the tuning fluctuations in the visual cortex[57,58].

### Higher neural activity barely increases, or even decreases Fisher information

So far, by applying our Pf-PCA analysis to the V1 data, we have derived a generative model of the neural activity in V1. Below, we demonstrate that this generative model is useful for characterizing various critical aspects of the neural code. In this section, we will leverage Pf-PCA to the calculation of the Fisher information to understand how the tuning fluctuation affects the information-carrying capacity of the V1 population. We focus on a local measure of the representation, i.e. Fisher information (FI), which has been important in quantifying the local property of the neural code[59–62]. In the next section, we will use Pf-PCA to understand how the geometry of the neural response changes under tuning fluctuations, which represents another important aspect of the neural code. Overall, through the FI and geometry analysis, Pf-PCA enables further understandings of the local and global structure of the V1 code.

First, using the model estimated by Pf-PCA, we examined the relationship between the FI and the magnitude of neural activity for individual neuron (Fig. 6a, b). See "Methods" for the calculation of the FI. We found that this relationship differs substantially from neuron to neuron- it could be positive, negative, or flat (Fig. 6a, b). Figure 6b shows the histogram of the slope when regressing FI against the neural activity. Interestingly, the median of these slopes is close to 0 (i.e., 0.001). We also reported the histogram of the slopes of FI-activity curve for each session in Supplementary Fig. 13. We further validated these results by performing recovery analysis using synthetic data. We found that our method could indeed faithfully recover the relationship between FI and neural activity for individual neurons given the particular sample size of the data (see Supplementary Fig. 14).

Figure 6c shows the population FI obtained by summing the FI values across all stimuli within a block and all neurons in each dataset, sorted according to the neural activity of individual blocks. Here we assume that the neurons are noise-independent conditioned on the latent fluctuations. Note that a multiplicative gain model predicts that the population FI scales proportionally with the amount of neural activity, or put it in another way, doubling the firing rate would double the population FI. However, we found that, for most sessions, the population FI is minimally affected, or decreases systematically as the neural activity increases. This is in sharp contrast with the multiplicative gain model. To quantify this, we defined a FI-modulation index (i.e., the slope of FI-activity curve). With the multiplicative gain model, this FI-modulation is exactly 1. In the data, the modulation indexes for all sessions are far smaller than 1 and in some cases negative (−0.17, −0.44, −0.07, 0.10, 0.12, 0.26, and 0.21, respectively).

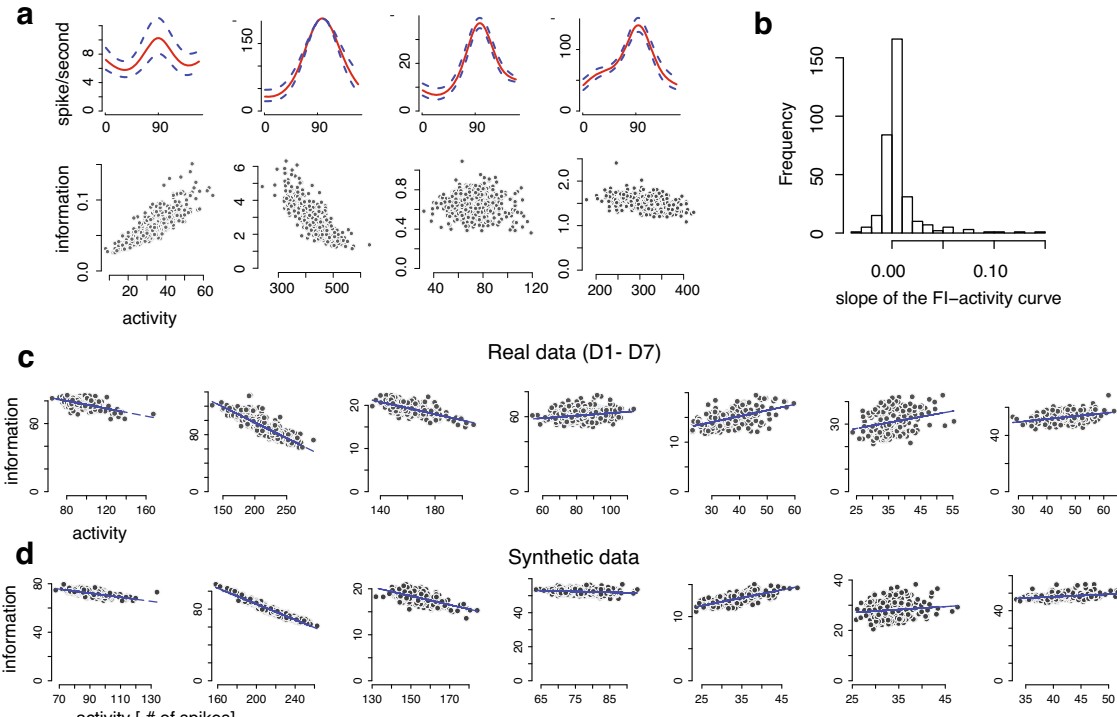

**Fig. 6 | Fisher information analysis reveals that higher neural activity barely increases, or even decreases Fisher information. a** The relation between the FI and the neural activity (lower panels) and the tuning curves with latent fluctuation induced by the first fPC (upper panels) in four neuron examples. **b** Histogram of slopes of the FI-activity for all neurons ($n = 402$). **c** Scatter plots showing the relationship between the population FI and the neural activity for each session in real data. **d** Scatter plots showing the relationship between the population FI and the neural activity for the synthetic data. The synthetic data were generated from the Pf-PCA model fit to each corresponding dataset. Note that there is a tendency for the model to underestimate the scores (as expected), leading to a proportional underestimation of the population FI. Importantly, this underestimation does not affect the recovery of the relation between the population FI and the neural activity.

Recovery analysis based on synthetic data suggests that our procedure could indeed recover the relationship between the neural activity and the population FI (Fig. 6d). See "Methods" for details. These results are consistent with the study[30] in that both studies found that increased neural activities do not lead to substantial increase of the population FI. Meanwhile, we also noticed some subtle discrepancies between the two studies, because it has been suggested[30] that there was a minimal change in the population FI when neural activity changes. We believe that the difference lies in the difference in the analysis methods (see "Methods" and Supplementary Note 5 for a detailed discussion).

**Change of representational geometry induced by spontaneous fluctuation is different from contrast**

Neural response naturally forms neural manifold by systematically varying the stimulus. The geometry of the encoding manifold has multiple implications in understanding the format of the representation and in linking neural responses to the behavior (reviewed in ref. 63). How the tuning variability affects the geometrical properties of the encoding manifold represents an interesting yet unresolved question. To investigate this question, we begin by simulating a multiplicative gain, which is a simple scenario exhibiting how fluctuating signals are displayed by the geometric analysis. We constructed a homogeneous population code for encoding stimulus orientation with independent Poisson noise, and varying the shared multiplicative gain (see "Methods" for details). This population coding model recapitulates the basic effect of varying stimulus contrast[22,23,44]. We computed the representational distance[64] as a function of the orientation disparity, and found that multiplicative gain only scales the representational distance function without changing the shape (Fig. 7c). A 3-D multi-dimensional scaling (MDS) based on the representational distance matrix shows

that the neural manifold under tuning fluctuation exhibits a cone-shape (Fig. 7d), with the radial dimension encoding the multiplicative gain. When projecting onto the first two dimensions, we observed that the size of the representation for each contrast (e.g., the radius of each circle) scales with the neural activity (Fig. 7e).

Next, we sought to understand how the tuning fluctuations identified by Pf-PCA from the V1 data would affect the geometry of the code[65,66]. To do so, we created a neural population code based on the empirically fitted tuning curves and scores from Pf-PCA (see "Methods" for details). We clustered the score matrix into 10 clusters, then computed the average scores for each cluster to get the pattern of fluctuations corresponding to each of the 10 characteristic states. For each state, the corresponding tuning curves were generated accordingly. Analyzing the representational distance (RD) as a function of orientation disparity for the 10 latent states (shown in Fig. 7g–m, first column), we found that the RD curves are only slightly affected by the total activity. In several cases, higher activity leads to overall lower RD, e.g., (Fig. 7h). Furthermore, MDS analysis (Fig. 7g–m, second column) shows that the fluctuations cause the representation to move along a cylinder-like manifold. When projecting onto the first two dimensions, we had two observations. First, the centers of the representations corresponding to all states are aligned, suggesting that the representation "drifts"[67] in the direction is orthogonal to the representation of orientation. Second, the size of the representation only changes slightly with varying population activities (Fig. 7g–m, third column). Note that this general pattern does not resemble the cone-like structure induced by the multiplicative gain (Fig. 7d). Note that in two of seven sessions, the latent fluctuations are smaller so that the cylinder structure does not appear in the 3-D MDS, however, it becomes apparent when we plotted the first two and the fifth dimension in a 5-D MDS embedding.

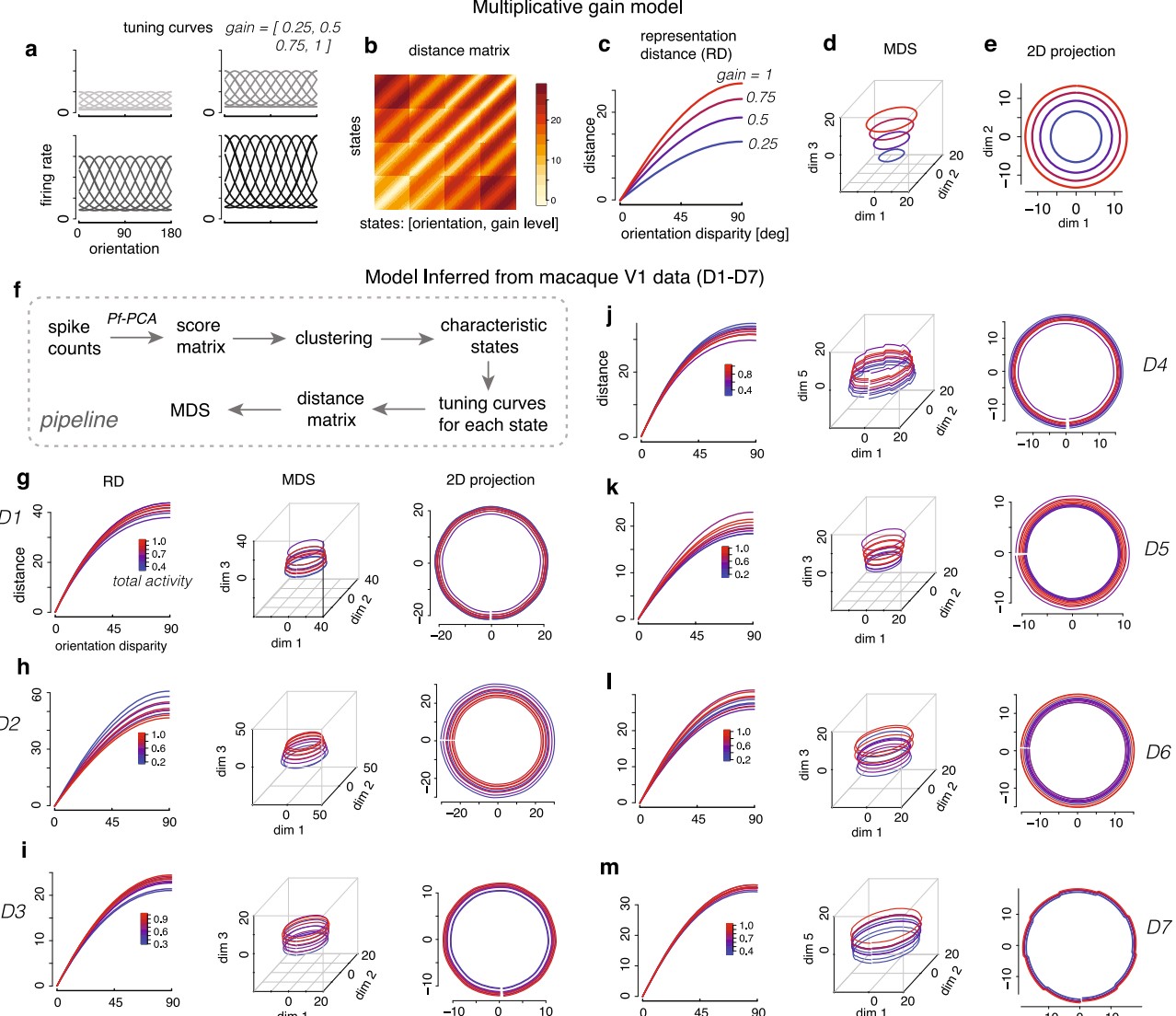

**Fig. 7 | Geometry analysis demonstrates that the latent tuning fluctuations lie on a different manifold compared to changing contrast. a–e** Analysis based on multiplicative gain model. **a** Tuning curves under different multiplicative gain (or contrast) levels. **b** The representational distance matrix for each pair of states, defined by the orientation and the gain level. We discretized the orientation into 180 bins, results in 180 × 4 = 720 states. The states were arranged according to the orientation and the gain level. **c** The representational distance (RD) as a function of orientation disparity for four different gain levels. **d** 3-D MDS revealing a cone-like structure of the neural population code. **e** Projection onto the first two dimensions revealing that the size of the representation (measured by radius) scales

proportionally with the neural activity. **f** The pipeline of geometric analysis based on real data. For each session, we inferred the score matrix for the first fPC, and then computed ten characteristic states using a clustering procedure. For each characteristic state, we generated the corresponding tuning curves using the results from Pf-PCA. The curves for different states are color-coded based on the total activity (normalized to the maximum). **g–m** The results for each dataset from the geometric analysis. The MDS results reveal a cylinder-like structure in most of the sessions. Although total activity can change substantially across the different scales, however, the size of the representation does not change substantially. This is apparent when comparing it to the multiplicative gain model.

These results demonstrate that the spontaneous fluctuations of neural tuning in V1 lie on a different manifold compared to that induced by the changing stimulus contrast that leads to a multiplicative gain. Note that the effect is also different from a simple additive modulation (Supplementary Fig. 16). These results have important implications for the downstream readout. If the spontaneous fluctuations lied on the same manifold as the changing stimulus contrast, the downstream would not be possible to distinguish the spontaneous fluctuations from a change of contrast. Our results argue against that scenario, and further suggest that the latent fluctuations mostly cause a "drift" of the representation without fundamentally changing the fidelity and the structure of the representation[67–69].

## Discussion

We have presented a flexible unsupervised approach, Pf-PCA, to analyze the tuning variability. This approach provides a general framework to understand how the observed stimulus variables and latent factors together influence the neural activity. Specifically, it decomposes the tuning curves as the sum of the mean component and the fPCs which are tuned to the stimulus, and are subject to the modulation of the latent factors. We demonstrated that Pf-PCA could robustly and reliably recover the structure of the fluctuations given a few dozens of blocks of data. We applied our method to analyze the spike train data collected from anesthetized macaque V1 while viewing drift gratings, and discovered several insights regarding the structure of the orientation code.

Our method represents a more flexible modeling framework compared to previous work in analyzing the tuning variability. Previous models often presumed the form of fluctuation (e.g. refs. 25,28,30,31), and the fluctuation was often assumed to be a constant, acting onto the tuning curve through either multiplicative or additive interactions. Thus, the forms of fluctuations captured by these analyses were limited by construction. Our method, instead, allows unsupervised discovery of arbitrary smooth tuning fluctuation, and potentially multiple forms of fluctuations simultaneously. Our method is broadly applicable, so long as the neurons have smooth tuning over certain stimulus dimension, which could be spatial frequency[70], location[17,71], direction[72,73], or time[74]. A potentially fruitful venue of using our approach would be to leverage our approach to test computational models by analyzing the data simulated from these models to identify the structure of the latent fluctuations, and comparing these predictions with the structure extracted from the data.

Our results suggest that the tuning fluctuation exhibits low-rank structure, both at the levels of individual neurons and neural populations. The latter is generally consistent with, and generalizes from, results in previous studies that assumed coherent gain fluctuations among simultaneously recorded neurons[26,28,29]. A recent study[75] found that the variance explained by the PCs of large-scale neural populations scaled as a power-law. Our results are different from theirs: (i) our results concern about the dimensionality of tuning variability, not the dimensionality of stimulus tuning; (ii) we primarily focus on the amount of variance explained by the top PCs, not the properties of the tail of the spectrum as done in ref. 75. With a few dozens of simultaneously recorded neurons, we can not accurately estimate the scaling relationship between the variance (of the tuning variability) explained of the neural population and the number of PCs—an interesting question that could be addressed in the future with larger datasets.

We have focused on an exponential non-linearity for the link function, which has been assumed by many previous models[76–79]. It should be possible to further extend it to other types of non-linearity[80], such as a power-law transformation[32,44,81]. It would also be interesting for future research to develop techniques that could automatically infer the type of non-linearity from the data directly.

Our V1 results should be informative to a better mechanistic and functional understanding of the V1. Naively, assuming an exponential non-linearity, the power-law modulation revealed by our analysis could be explained by a tuned input to a given neuron which fluctuates over time. However, this is likely a simplified picture. It would be more fruitful to consider how threshold non-linearity together with noise could lead to these kind of results. Previously, models of this kind[81–83] have been used to account for the multiplicative gain on the tuning curve induced by varying contrast. Second, the finding that the latent fluctuations are heterogeneous in the population is consistent with the idea that the recurrent processing in V1 may play an important rule in shaping the structure of the fluctuations of neural tuning. These results echo with recent work[57] showing that spatially patterned fluctuation structure could emerge in balance networks in V1 in which neural fluctuations can be heterogeneous. Third, it is interesting to consider the implications of our observations in the context of the functional models of neural variability. Such variability has been proposed to reflect sampling of the sensory inputs[84], encoding stimulus uncertainty[27], and efficient encoding of natural scene statistics[85–87]. The specific structure of the latent fluctuations extracted by Pf-PCA provides a richer set of summary statistics to further test these current mechanistic and functional models and help developing future models.

Our method enables us to further analyze the coding properties in the presence of the tuning fluctuations, both in terms of local properties (via FI) and the global geometrical structure of the code under tuning fluctuations. We found that FI generally does not substantially increase (sometimes even decrease) with increased neural activity. This may point to the potential importance of cortical inhibition in sharpening the neural code[88]. The analysis of the geometry reveals that the manifold induced by the latent fluctuation lies in different subspace of changing contrast. This suggests that the tuning fluctuations in V1 may not interfere with encoding of contrast. These observations deserve further investigations in the future.

Our V1 results are entirely based on analyzing neural responses at the anesthetized state. To the extent by which the structure of noise fluctuations under the anesthetized states resembles those of the awake-behaving animals remains an open problem. Earlier work using voltage-sensitive dye to measure large-scale activity fluctuation in V1 under anesthesia found that the structure of the spontaneous fluctuations resembled the stimulus-driven activity, and they interacted with stimulus-evoked activity in an additive fashion[5,25,89], and reported that slow gain fluctuations identified in the anesthetized macaque were also present in the awake state. In addition, results in ref. 30 found that the additive and multiplicative change of the tuning curves were also present in a smaller dataset from one macaque monkey. Nonetheless, anesthesia can change the properties of neuron integration in cortical neurons[90], and may trigger a profound change of the cortical dynamics[91] and coding[92]. Detailed in-depth investigations in the future will be important to determine whether the rule of V1 tuning variability that we discovered from anesthetized states may generalize to the awake states. One further limitation of the anesthetized data is that it precludes the analysis of the latent tuning fluctuations with behavior. It would be interesting to see if the fluctuations of internal states similar to what we found here correspond to a change of the behavior[69].

A few limitations and the potential improvement of our results are worth mentioning. Our current method does not explicitly model the temporal structure of the tuning fluctuation, as $\alpha_t$ is assumed to be independent, and is estimated for each $t$. It should be possible to improve our method by leveraging temporal smoothness prior on the scores, e.g., by weighted Poisson fPCA where the weights are constructed by using a temporal kernel, or assuming a Gaussian process prior[93–98]—a direction we did not pursue here, but would be an interesting future direction. Also, our method, when applying to V1 data, only deals with slow fluctuations (~10 s)[25] (see Supplementary Fig. 8), because of the assumption that the latent is the same within every block (or moment). Thus, this inferred moment-to-moment tuning fluctuation is at the time scale of ~10 s. Tuning variability at an even faster time scale would be averaged out. Thus our estimate of the tuning fluctuation is likely an under-estimate of the true fluctuations. It should be possible to refine these estimates and study tuning variability at an even faster scale. Two approaches may be promising: (i) by using faster stimulus sampling in experiments—with a stimulus sampling of 100 ms per stimulus, it is possible to apply the same approach to study tuning variability at a timescale of ~1 s; (ii) by extending our methods to fit the neural population all together. For the latter, assuming a low dimensional latent structure, it should be possible to infer the latent fluctuation based on individual stimulus—a direction we are currently pursuing.

In summary, we have developed a statistical approach to parse the variability of neural tuning. Our approach can flexibly capture the impact of both stimulus variable and latent variable on a moment-by-moment basis. Applying our approach to macaque V1 revealed the structure of the tuning variability both at the level of individual neuron and the neural population. Our analyses also led to further insights on the FI and geometry of the code. While we only analyzed the orientation code for V1 in the paper, we hope that the analysis pipeline developed here would be informative for elucidating the structure of neural tuning and response variability in other neural systems as well[99].

## Methods

### Poisson functional principal component analysis (Pf-PCA): generative model

A standard model description of neural responses in neuroscience is based on tuning curves and (typically) Poisson spiking noise.

Specifically, the observed spike count $n(s)$ of stimulus $s$ for individual neuron, during a counting window of length $\Delta_t$, is modeled as a Poisson distribution

$$p(n(s)|\mu(s)\Delta_t) = \frac{(\mu(s)\Delta_t)^{n(s)}}{n(s)!}\exp(-\mu(s)\Delta_t). \qquad (4)$$

where $\mu(s)$ represents the tuning curve of stimulus $s$.

The tuning curve may vary among $B$ moments (i.e., blocks of trials) and is not directly observed. Denoting the tuning curve as $\mu_t(s)$ for moment $t$, $t = 1, \cdots B$, we model the log of the stochastic curve, $\log(\mu_t(s))$, as following:

$$\log(\mu_t(s)) = f(s) + \sum_k \alpha_{k,t}\phi_k(s) + \epsilon_t(s). \qquad (5)$$

Here $f(s)$ is the mean component, $\phi_k(s)$ is the $k$-th functional principal component (fPC), $\alpha_{k,t}$ denotes the amount of fluctuation (i.e., score) for the $k$-th component during the $t$-th moment, and is assumed to follow a zero-mean Gaussian distribution, with variance $\sigma_k^2$, and $\{\epsilon_t(s)\}$ are independent and identically distributed zero-mean Gaussian noise with variance $\sigma_0^2$ that is a parameter to quantify the remaining variance of $\log(\mu_t(s))$ on $\{\phi_k(s)\}$. Note that this is the same model as described in Eq. (1) in the main text, in which dependence of the individual terms on $s$ was suppressed for simplicity. This model implies that $\log(\mu_t(s))$ has the mean $E[\log(\mu_t(s))] = f(s)$ and the co-variance function $\sum_k \sigma_k^2 \phi_k(s_1)\phi_k(s_2) + \sigma_0^2$.

Note that with only the first term $f(s)$, this model is equivalent to the standard tuning curve model of spike counts. The second term $\sum_k \alpha_{k,t}\phi_k(s)$ is the additional variance from the contribution of moment-to-moment fluctuations, making the model naturally capture the overdispersion of spike counts.

Pf-PCA differs from the multiplicative gain model proposed in ref. 25. First and most importantly, we do not assume a pure multiplicative gain change in the fluctuations, instead, the form of fluctuations is arbitrary. Second and a more subtle point is that, in Pf-PCA, the magnitude of the fluctuation $\alpha$ is assumed to follow a Gaussian distribution, while Gamma distribution was assumed for the gain in the firing rate scale (not the logarithm of firing rate) in ref. 25. Note that $\alpha_{k,t}$ is Gaussian distributed thus symmetric, while the logarithm of Gamma distribution is left-skewed thus has a different shape.

## Inference: a two-step estimation procedure

Assume that we have observations based on a set of $m$ stimuli which are sampled from a particular stimulus space $\mathcal{S}$. The spike count of individual neuron elicited by individual stimulus $s_j$, $j = 1, \ldots, m$, is denoted as $n_t(s_j)$ for the $t$-th moment (i.e., the $t$-th block of trials). We further denote the spike count vector for the $t$-th moment as $\vec{n}_t = (n_t(s_1), \cdots, n_t(s_m))^\top$.

Intuitively, if we could recover the unobservable mean $\mu_t(s_j)$ for $t = 1, \ldots, B$, $j = 1, \ldots, m$, fitting the model of stochastic curve $\log(\mu_t(s))$ using functional PCA would be straight-forward. Denote $\vec{\mu}_t = (\mu_t(s_1), \cdots, \mu_t(s_m))^\top$. We could estimate the posterior of the hidden firing rate $\log(\vec{\mu}_t)$ from the spike count data using an expectation-maximization (EM) algorithm[100]. Following these ideas, we developed a two-step estimation procedure, as follows.

**Step 1: recover the hidden** $\vec{\mu}_t$

When the vector $\log(\vec{\mu}_t)$ is observable, the likelihood of Poisson model can be written as

$$\ell_p = \sum_t \sum_j \left\{ \frac{[\mu_t(s_j)]^{n_t(s_j)}}{\mu_t(s_j)!} \exp\left[-\mu_t(s_j)\right] \right\}. \qquad (6)$$

Eqn. (5) implies that the logarithm of the firing rate for the sampled stimulus set $\{s_1, \cdots, s_m\}$ during the $t$-th moment $\log(\vec{\mu}_t)$ can be

modeled as $\log(\vec{\mu}_t) \sim N(\vec{f}, \Sigma)$, where $\Sigma = \sum_k \sigma_k^2 \vec{\phi}_k \vec{\phi}_k^\top + \sigma_0^2 \mathbf{I}$. We then obtain the log-likelihood

$$\ell_g = \sum_t \left( \log(\vec{\mu}_t) - \vec{f} \right)^\top \Sigma^{-1} \left( \log(\vec{\mu}_t) - \vec{f} \right) + \log|\Sigma|. \qquad (7)$$

In reality the firing rates are not directly observed. However, by treating $\log(\vec{\mu}_t)$ as missing data, we could use an EM algorithm to iterate between an E-step and a M-step to optimize the functions. Specifically, the E-step calculates the conditional mean $E[\log(\vec{\mu}_t)|\vec{n}_t]$ and conditional variance $Cov[\log(\vec{\mu}_t)|\vec{n}_t]$, given the current estimates of the parameters $\vec{f}$ and $\hat{\Sigma}$ obtained in M-step. Given the expectations obtained in the E-step, the M-step maximizes $\ell_g$ in Eq. (7), which involves the following two quantities:

$$\hat{\vec{f}} = \frac{1}{B}\sum_{t=1}^B E[\log(\vec{\mu}_t)|\vec{n}_t],$$

$$\hat{\Sigma} = \frac{1}{B}\sum_{t=1}^B \left[ Cov[\log(\vec{\mu}_t)|\vec{n}_t] + \left[ E[\log(\vec{\mu}_t)|\vec{n}_t] - \hat{\vec{f}} \right]\left[ E[\log(\vec{\mu}_t)|\vec{n}_t] - \hat{\vec{f}} \right]^\top \right].$$

Note that calculating these expectations in the E-step requires the marginal distribution $\vec{n}_t$, which is not analytically tractable. We thus adopt a Monte Carlo approach to calculate them. For each $t$, we generate a set of samples, $\log(\vec{\mu}_t)^{*1}, \cdots, \log(\vec{\mu}_t)^{*M}$, where $M = 10,000$ is the number of Monte Carlo runs, according to the distribution of $\log(\vec{\mu}_t)$ given by the current parameters. Then the unbiased estimates are obtained from the samples that we simulated. Together, **Step 1** gives an estimator of hidden means $\vec{\mu}_t$ in the form of $E[\log(\vec{\mu}_t)|\vec{n}_t]$.

**Step 2: perform functional PCA on the recovered hidden** $E[\log(\vec{\mu}_t)|\vec{n}_t]$

Given the posterior $E[\log(\vec{\mu}_t)|\vec{n}_t]$, we then apply the method of functional data analysis into the estimated posterior means to get the mean component $f(s)$, the fPCs $\{\phi_k(s)\}$, and their corresponding scores $\alpha_{k,t}$. Specifically, $f(s)$ is obtained by using the natural cubic splines smoothing approach

$$\ell_s = \sum_{t=1}^B |E[\log(\vec{\mu}_t)|\vec{n}_t] - \vec{f}|^2 + \lambda \int \left( \frac{\partial^2 f(s)}{\partial s^2} \right)^2 ds,$$

where $\lambda$ is chosen via generalized cross validation[101].

The functional fluctuations $\{\phi_k(s)\}$ are estimated by the roughness of the eigenfunction[52,102]. The first component $\vec{\phi}_1$ and the corresponding score for each moment $\hat{\alpha}_{1,t}$ are estimated via

$$\max_{\phi_1(s)} \frac{\text{var} \int \phi_1(s)E[\log(\vec{\mu}_t)|\vec{n}_t]ds}{1 + \lambda \int (\phi_1''(s))^2 ds} \text{ subject to } \int (\phi_1(s))^2 ds = 1,$$

$$\text{and } \hat{\alpha}_{1,t} = \int \phi_1(s)[E[\log(\vec{\mu}_t)|\vec{n}_t] - f(s)]ds. \qquad (8)$$

The remaining components and their scores are obtained via an iterative process such that any higher order eigenfunction is orthogonal to the eigenfunctions already recovered. This procedure allows us to estimate the variance explained by each fPC, as quantified by the variance of the score for that component. The proportion of variance explained by the $k$-th fPC $\vec{\phi}_k$ is simply calculated by $\text{var}(\hat{\alpha}_{k,t})/\sum_{k'}\text{var}(\hat{\alpha}_{k',t})$.

## Implementation of a reduced version of the method: $\mu$-PCA

We also implement a reduced version of the method, $\mu$-PCA. For this method, after obtaining the posterior mean $E(\log(\vec{\mu}_t)|\vec{n}_t)$, we perform regular PCA directly on the exponential function of the estimated posterior mean, $\exp[E(\log(\vec{\mu}_t)|\vec{n}_t)]$, instead of functional PCA,

$\mu$-PCA can be thought as an alternative way to perform PCA to Poisson count data compared to Poisson PCA[54]. First, Poisson PCA[54]

considered $\log(\mu_t(s))$ as "natural parameters", which define the mean and components. In contrast, $\mu$-PCA considers the unobserved $\log(\mu_t(s))$ as random variables, and decomposes it into the mean and the principal components with the amount of fluctuations assumed to be Gaussian. Second, different techniques are proposed to obtain the principal components. Following the relation between log-likelihood of exponential family and Bregman distance, Poisson PCA[54] constructed a loss function to optimize it, and then obtained the principal components. The $\mu$-PCA method seeks to estimate the posterior mean $E(\log(\vec{\mu}_t)|\vec{n}_t)$ instead of random variables $\vec{\mu}_t$.

### Validation of the methods using simulated data

We validated our methods using simulated data inspired by experimental observations[30,55,103] and with ground truth. To do so, we generated tuning curve $\mu(s) = 0.5 + \frac{5}{f_G(0)} f_G(\frac{s}{20})$ where $f_G$ is the density function of standard normal distribution. The stimulus $s$ takes a sequence with a 22.5 degree interval from −90 to 90 degree. The number of moments (blocks) $B$ was set to be 50 throughout.

For each of the four types of tuning changes, i.e., multiplicative gain, additive modulation, tuning shift, and tuning sharpening, we reverse-engineered the fPC $\vec{\phi}$ that would give rise to that type of tuning fluctuations. More concretely, the tuning fluctuations, denoted by $\vec{\gamma}_{t}$, are generated from Gaussian with covariance structure $\Sigma = \sigma_1^2 \vec{\phi}\,\vec{\phi}^\top + c\mathbf{I}$, where $c$ is chosen such that the structured component explains 80% of the variance. The second component amounts to white noise, accounting for 20% of the variance. Adding this random noise component allows us to test the robustness of the model in the presence of less structured fluctuations, as well as to evaluate the extent to which our estimation procedure could recover the variance explained by the structured fPC. Note that the recovery problem becomes easier without such random noise component. From the simulated data, we first estimated the variance explained by the $K$ fPCs ($K$ = number of stimuli) using Pf-PCA, and then computed the proportion of variance explained by the first fPC as $\mathrm{var}(\hat{\alpha}_{1,t})/\sum_{k'}\mathrm{var}(\hat{\alpha}_{k',t})$. Effectively, we relied on the last $K - 1$ components to recover the random component in the generative model.

Here $\mu_0(s) = \exp(f(s))$ represents the tuning curve with no fluctuation, and $\sigma_1^2$ denotes the the strength of fluctuation. Due to the scale difference among different fluctuations, the parameter $\sigma_1^2$ is set as follows. Multiplicative gain: $\sigma_1^2 = 1.25$, so that its gain change is $\log(1.3\mu_0(s)) - \log(0.9\mu_0(s))$. Additive modulation: $\sigma_1^2 = 5.5$, so that its response change is $\log[\mu_0(s) + 0.4] - \log[\mu_0(s) - 0.2]$. Tuning shift: $\sigma_1^2 = 1.38$, so that it shifts $\log(\mu_0(s+6)) - \log(\mu_0(s-6))$. Tuning width change: $\sigma_1^2 = 1.85$, so that its standard deviance changes ± 20%. Under these settings, we generated Poisson spike count data with firing rates $\mu_t(s) = \exp[\log(\mu_0(s)) + \gamma_t(s)]$.

Note that we also validated our methods (i) when multiple forms of fluctuations co-exist simultaneously; (ii) when the tuning curves are monotonic[104,105]. These procedures and their results can be found in Supplementary Notes 2 and 3.

### Analyzing data from macaque V1 using Pf-PCA

We used 7 datasets (i.e., 7 sessions) which were published previously. Three of them (D5-D7), publicly available from CRCNS website, were obtained from anesthetized macaque primary visual cortex by Matthew Smith and Adam Kohn[55]. In these experiments, described in details in refs. [55,103], spiking activities were recorded while presenting different grayscale visual stimuli, including drifting sinusoidal gratings (each presented for 1.28 s). These gratings are [0, 30, 60, ⋯, 330] deg. The other four sessions (D1-D4) were previously published in[30], shared by the authors. These data also visually evoked activities from anesthetized macaque primary visual cortex (see ref. [30] for details). The grating directions are [0, 22.5, 45, ⋯, 157.5] deg. We chose neurons

with SNR≥2 and mean firing rate ≥1.5 spikes/second. In total, we analyzed 7 datasets with 402 neurons.

For each stimulus, we counted the number of spikes for a 500 ms window (80–580 ms after stimulus onset). Because the experiments had a block-randomized design, for each block we obtained a response vector corresponding the responses for all the stimulus orientations sampled in the experiments. Repeating this for every block, we constructed a spike count matrix for each neuron (number of blocks × number of orientations).

We then applied Pf-PCA to this matrix for each neuron. In doing so, we obtained the mean component $f(s)$, the fPCs $\phi_k(s)$, where $k$ is component index, as well as the amount of fluctuations, i.e., scores, $\alpha_{k,t}$ for each moment $t$. We set the number of fPCs to be to three, as three fPCs could already sufficient to account for most of the variance (see Supplementary Fig. 6). The results reported in Fig. 3 were obtained based on hundreds of blocks of data (400 for the D1-D3, 200 for the D5-D7). To examine the impact of sample size, we ran Pf-PCA on subsets of V1 data by taking 25, 50, or 100 blocks of each dataset. See the results in Supplementary Note 4 and Supplementary Fig. 7.

In Fig. 3a, we plotted the inferred mean component in the form of $\exp(f(s))$, and the first fPC in the form of $\exp(f(s) \pm \sigma\phi_1(s))$, where $\sigma$ is the s.d. of estimated $\alpha_{1,t}$.

**Regression analysis.** We analyzed the relationship between $\phi_1(s)$ and $f(s)$ by performing a regression analysis with the following form

$$\phi_1(s) = b + wf(s) + e(s). \tag{9}$$

The regression is done by using the "lm" function in scientific computing software R. First, we tested the significance of the regression by a F-test (Fig. 3c). To quantify how much information of $\phi_1(s)$ is accounted for by a linear function of $f(s)$, we defined a summary statistics: $\texttt{fraction} = 1 - \frac{\sum_s e(s)^2}{\sum_s \phi_1(s)^2}$. This measure was reported in Fig. 3d.

**"Flatness index" analysis.** In Fig. 4a, $\mu_0(s) = \exp(f(s))$ was generated from von Mises function with parameters satisfying $\mu_0(s) \in [0.2, 1]$. Denote the tuning curve corresponding to the fluctuation $\alpha = \alpha_0$ as $\mu_{\alpha_0}(s)$. Define $\Delta\mu(s) = \mu_\alpha(s) - \mu_0(s) - c(\exp(b\alpha) - 1)$, where $c$ is the baseline of $\mu_0(s)$. Thus, $\Delta\mu(s)$ captures the change of firing rate as a function of the stimulus with an additional correction term. The "flatness" index was defined as $\frac{\Delta\mu(s^{orth})}{\Delta\mu(s^{pref})}$, where $s^{pref}$ and $s^{orth}$ denote the preferred orientation of the neuron and its orthogonal orientation, respectively.

This index quantifies how flat $\Delta\mu(s)$ is. For additive change, $\mu_\alpha(s) = \mu_0(s) + c_{add}\alpha$, where $c_{add}$ is a constant, and $\Delta\mu(s) = c_{add}\alpha - c(\exp(b\alpha) - 1)$, implying $\Delta\mu(s)$ is completely flat over $s$. Thus, $\texttt{flatness} = 1$ in this case. For multiplicative gain, $\mu_\alpha(s) = \exp(b\alpha)\mu_0(s)$ and $\Delta\mu(s) = (\mu_0(s) - c)(\exp(b\alpha) - 1)$, implying $\Delta\mu(s^{orth}) = 0$. Thus, $\texttt{flatness} = 0$. Remarks: $\texttt{flatness}$ can be larger than 1 when $\Delta\mu(s^{orth}) > \Delta\mu(s^{pref})$, which is possible when $\Delta\mu(s) < 0$. It is also possible to have $\texttt{flatness} < 0$, when $\Delta\mu(s^{orth}) < 0 < \Delta\mu(s^{pref})$.

**Connecting the power-law relation to multiplicative gain and additive modulation.** The power-law modulation can degenerate to multiplicative gain and additive modulation under certain parameter regime. Obviously, as $w = 0$, the power-law modulation is equivalent to multiplicative gain. The connection to additive modulation is less obvious. When $b\alpha$ and $w\alpha$ are close to 0, using Taylor expansion, $\mu_\alpha(s) \approx \mu_0(s) + \mu_0(s)(b + w\log\mu_0(s))\alpha$. It follows that when the function $\mu_0(s)b/w + \mu_0(s)\log\mu_0(s)$ is flat over $s$, the power-law modulation degenerates to an additive change. Examining the property of the function $g(x) = x\log(x) - \kappa x$, we found that $g(x)$ is indeed

approximately flat over $[0, 180]$ when $\kappa$ is in some region, resulting in an approximate additive modulation.

## Control analysis

**Simulated data from multiplicative gain model.** To generate synthetic data from the multiplicative gain model, we set the tuning curve $\mu(s)$ on each moment to be log(mean firing rate) plus a constant fluctuation, with its standard deviation matching that inferred from the real data. We then sampled the spike count under Poisson noise. In doing so, we generated synthetic data which approximately match the amount of fluctuations in the real data, but with a pure multiplicative gain. We performed regression analysis on the simulated data using the same procedure as that was used for real data (described above). The slope values obtained for the real and synthetic data were compared. The results were reported in Fig. 3e.

**Rank-1 model.** In Fig. 5c, d, f and Supplementary Fig. 11, we reported the recovered score matrix based on a Rank-1 multiplicative gain model and the dimensionality. When simulating this model to generate synthetic data, the multiplicative gain fluctuations were sampled i.i.d. from a normal distribution. To ensure comparability of results, we set the variance in a way such that the fluctuations of each simulated neuron match the values of real neural data. The fluctuation at each moment was shared by all neurons in the population, ensuring the score matrix is rank-1. We performed the Pf-PCA analysis on the simulated population, then obtained the recovered score matrix. We also simulated and analyzed synthetic data from a rank-1 additive-modulation model. These results were reported in Supplementary Fig. 12.

## Fisher information

Assume that the tuning curve for neuron $i$ is $\mu^{(i)}(s)$ and that the spike count $n^{(i)}(s)$ follows Poisson distribution with mean $\mu^{(i)}(s)$. Because we can approximate $\mu^{(i)}(s)$ by the mean $f(s)$ plus the functional fluctuations, the Fisher information (FI) of neuron $i$ at stimulus $s$, given the score $\alpha_k$, where $k$ is index of component, is obtained by

$$
\begin{aligned}
I^{(i)}(s) &= \frac{\partial \mu^{(i)}(s)}{\partial s} E\left[ -\frac{\partial^2 [n^{(i)} \log(\mu^{(i)}) - \mu^{(i)}]}{(\partial \mu^{(i)})^2} \right] \frac{\partial \mu^{(i)}(s)}{\partial s} \\
&= \left[ \mu^{(i)}(s) \left( \frac{\partial f^{(i)}(s)}{\partial s} + \sum_k \alpha_k \frac{\partial \phi_k^{(i)}(s)}{\partial s} \right) \right] \frac{1}{\mu^{(i)}(s)} \left[ \mu^{(i)} \left( \frac{\partial f^{(i)}(s)}{\partial s} + \sum_k \alpha_k \frac{\partial \phi_k^{(i)}(s)}{\partial s} \right) \right] \\
&= \mu^{(i)}(s) \left[ \frac{\partial f^{(i)}(s)}{\partial s} + \sum_k \alpha_k \frac{\partial \phi_k^{(i)}(s)}{\partial s} \right]^2 .
\end{aligned}
\tag{10}
$$

To compute the population Fisher information, we assumed that the neurons are independent conditioned on the fluctuations. To compute the population FI for each stimulus, we summed over the neurons in the population. Note that the reported FI for neural population or individual neurons (Supplementary Fig. 14) is the total FI (Fig. 6) by summing over different stimulus orientations with an individual experimental block.

**Recovery analysis on FI.** To see if our method indeed has the statistical accuracy to recover the relation between FI and spike activity, we performed a control recovery analysis. We first generated synthetic datasets by simulating data based on the Pf-PCA model with the parameter values estimated from the real data. Specifically, for a neuron we considered the Poisson mean $\mu_t(s)$ of moment $t$ to be $\log(\mu_t(s)) = f(s) + \alpha_{1,t} \phi_1(s)$, and generated the counts of moment $t$ from Poisson with the mean. From this, we obtained the synthetic population counts. We then performed the same analysis pipeline on these synthetic data to estimate the population FI. From this control analysis, we found that our method can accurately recover the relationship between FI and total spiking activity.

**FI and classification analysis.** We performed classification analysis similar to ref. 30 to examine the relation between the population FI and classification accuracy. Similar to ref. 30, we split the data into two groups (i.e., high and low), sorted by the population activity. We performed classification based on ensemble with different size. Given a randomly selected ensemble of neurons with certain size, we performed multinomial logistic regression, and obtained the performance (proportion of correct classification). For avoiding over fitting the data, we used 5-fold cross-validation and reported the average performance across the five sets of left-out data. For each ensemble size, we performed this analysis on 500 randomly selective groups for high and low group each. These results were reported in Supplementary Fig. 15a. We also performed this analysis on the synthetic data as described above. The results were shown in Supplementary Fig. 15b.

## Analysis of representational geometry

We analyzed the geometry of the representation under a simple multiplicative gain model and the power-law model inferred from the V1 data.

For the simple multiplicative gain model recapitulating the effect of changing contrast, we generated a homogeneous set of tuning curves using von Mises function (tuning width parameter equals 1). We assumed that the multiplicative gain modulated the firing rate of all neurons in the same way. In Fig. 7a–e, we assumed that the multiplicative gain could take four different levels (0.25, 0.5, 0.75,1), and computed the representation distance matrix by evaluating the representational distance for each pair of states (defined by both stimulus orientation and multiplicative gain). We performed three-dimensional classic MDS to visualize the geometrical structure of the representation, and obtained the projection onto the first two dimensions. A similar analysis was performed based on pure additive change (for results, see Supplementary Fig. 16).

For the models based on Pf-PCA inferred from real data (Fig. 7f–l), we performed the geometry analysis by the following steps:

(i) We first generated the mean firing rates for each moment $t$ by from our power-law modulation model. We clustered the blocks × neuron score matrix into 10 clusters according to blocks by k-mean, then computed the average score within clusters to get the "10-state averaged score", which is 10 × neuron matrix. For each of the 10 states in the population, the corresponding tuning curves were generated.

(ii) To reduce the biased sampling of neurons, we created a more shift-invariant neural population code by shifting tuning curves 8 times with 20 degree each. Our assumption here is that the neural code for orientation in V1 is roughly shift-invariant.

(iii) We calculated the euclidean distances between stimuli based on the extended population matrix (after variance-stabilizing transformation for Poisson noise, i.e., taking the square root transformation) to obtain a distance matrix, and performed the classic MDS based on this distance matrix. In most of the sessions, we performed 3-D MDS. In two of seven sessions, the latent fluctuations are smaller so that the cylinder structure does not appear in 3-D MDS. For these two sessions, we performed 5-D MDS. When plotting the first two and the fifth dimension in a 5-D MDS embedding, the cylinder-like structure is apparent.

## Reporting summary

Further information on research design is available in the Nature Portfolio Reporting Summary linked to this article.

## Data availability

No experimental datasets were collected in this study. Three of the seven datasets used in this study are available from CRCNS data sharing website. The remaining 4 datasets were originally collected in

Dr. Adam Kohn's lab[30]. Request of these datasets should be directed to the original authors who collected these data. Source data are provided with this paper.

## Code availability

The R code that implements the Poisson functional PCA method and related analyses is available in a public repository (GitHub: https://github.com/rong-zhu/PfPCA).

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

## Acknowledgements

We thank Adam Kohn, Matt Smith, and Inigo Arandia-Romero for sharing the V1 data. We thank Liam Paninski, Robbe Goris and Nikolaus Kriegeskorte for fruitful discussions. We thank Kenneth Kay, Yvonne Li, Matthew Whiteway, and Mattia Rigotti for comments on an earlier version of this paper. R.J.B.Z. acknowledges support from Science and Technology Innovation 2030 - Brain Science and Brain-Inspired Intelligence Project (2021ZD0200204), Shanghai Municipal Science and Technology Major Project (2018SHZDZX01), and National Natural Science Foundation of China (11871459). X.X.W. is supported by the startup funds provided by The University of Texas at Austin.

## Author contributions

R.J.B.Z. and X.X.W. jointly designed and performed the research, interpreted the results, and wrote the paper.

## Competing interests

The authors declare no competing interests.
