## [Peer Review File · Nature Communications]

Unsupervised Approach to Decomposing Neural Tuning VariabilityREVIEWER COMMENTS

Reviewer #1 (Remarks to the Author):

Zhu and Wei's manuscript title 'unsupervised approach to decomposing neural tuning variability' seeks a hidden structure of variability within visual neural responses to grating stimuli. They used 7 datasets recorded from anesthetized monkeys. They used a method similar to exponential-family PCA [Collins et al. 2002]. Main differences are independent Gaussian prior on the coefficients, expectation-maximization inference procedure, and cubic spline smoothness regularization. They found that more than 60% of the trial-to-trial tuning curve variability that exceeded the Poisson variability was captured by 1 component. Furthermore, they found that this component was correlated with mean tuning curve. Therefore, they propose a new model of tuning variability which is a Poisson tuning with power-function nonlinearity and 1-dimensional latent fluctuation. However, at a population level, this 1-d fluctuation per neuron were not synchronized, and formed a 1-3 dimensional structure. Two additional analyses, the Fisher information and the representational distance, that showed that the power-law exponents are not consistent with multiplicative model.

The method is straightforward and the story of the paper is presented well. I enjoyed reading the paper. The basic results are impressive and relevant. I'm glad to see a nonparametric approach to the tuning variability analysis.

Please make the analysis and plotting code available. It would help with reproducibility and specifying some of the implementation details that are omitted in the methods.

Fig 1b is confusing. Shouldn't the mean be a sharp quadratic that reaches extreme negative numbers quickly?

Sec 4.1 eq (5) is missing an overall noise term that corresponds to the σ_0^2 below eq (6)?

I'm unfamiliar with [30], and the population FI analysis didn't make much sense to me. I would appreciate if the authors can add why population FI as a function of firing rate is important (other than to show that multiplicative model's predictions do not hold; that point has been made already, right?)

I'm also unfamiliar with [62], and the representational distance, but I only really got the message that the tuning fluctuations and the stimulus contrast prediction which is multiplicative are not consistent.

Since the power-law exponents already showed that many of the neurons aren't multiplicative, this isn't surprising, right? I may be missing something.

The paper lacks in-depth discussion on anesthetized state. Although the authors indicated that it is a limitation of the current study, description on specifically how it could be limited would be useful.

As the authors noted in the discussion, since spike counts within a (~10 sec) block were considered to come from a particular random tuning curve, the fluctuations that are faster (which are typical in anesthesia) are lumped together within the nonlinear model.

This is more of a comment than a critique. No action is needed: Would it be possible to re-bin the data at a smaller temporal bin (e.g. 100 or 500 ms) and repeat the analysis? Would it change any of the conclusions? I understand the smoothness of tuning function would be more difficult to impose in that case. It is probably more appropriate to assume smoothness of the latent variable over time which causes the variability of the tuning curves. As the authors mention, there are many GP based factor models that would be able to tackle exactly this problem. I would also love to see the temporal structure of the fluctuations of the latent. But that's perhaps outside the scope of this paper.

Minor

- I think 'gain' to be multiplicative, so 'additive gain' is strange to me. I would prefer 'additive modulation', or simply 'additive'.

- Fig 3, are the 7 "sessions" the 7 "datasets"?

- Fig 3 caption, linear on the mean -> scaled?? (rephrase)

- Fig 3 caption, closed -> close to

- Fig 5 caption: bock -> block

- p12, can not -> cannot

- p13, is simple -> is a simple

- p16, what decoder is assumed for the "drift" of representation?

Reviewer #2 (Remarks to the Author):

This paper describes a new analysis of response variability of single neurons in visual cortex. Specifically, it uses a latent factor model to characterize slow changes in a neuron's tuning curve over time in terms of a small number of tuning-curve-like components modulated by Gaussian noise (and then passed through a nonlinearity). The resulting model can identify variable firing patterns over time as resulting from shifts in a tuning curve, tuning curve width changes, and additive or multiplicative response noise. The authors also provide novel analyses of the coding implications of this form of variability using Fisher information. I think the proposed framework interesting and important, likely to be of interest to a large segment of the wider field. However, there are two major shortcomings that should be addressed before the paper is suitable for publication.

1. The first problem is conceptual. The phrase "trial-to-trial tuning variability" (which appears in the abstract) is non-sensical (at least as I read it) given the definition of tuning curve, which is that it is the average or expected response to a stimulus. By definition, the tuning curve cannot fluctuate from trial to trial. The tuning curve IS the average response across trials, and the trial-to-trial fluctuations are (commonly known as) noise (NOT tuning).

At first I thought this was merely a terminology problem, but I grew more alarmed when I got to equation 1, which describes a factor model for single-neuron responses. This appears even more deeply nonsensical, since a single neuron responds only to the single stimulus presented on each trial. We can't model a vector of responses from a single neuron to multiple stimuli on a single trial! This makes no sense. A neuron can only respond to one stimulus at a time, so this seems tantamount to attempting to model noise correlations between neurons that were not recorded simultaneously.

It was only when I finally got to the Methods section of the paper that I realized that authors have something very different in mind when they refer to "a trial":

> pg 25: "Because the experiments had a block-randomized design, for each block we obtained a response vector corresponding the responses for all the stimulus orientations sampled in the experiments. Repeating this for every block, we constructed a spike count matrix for each neuron."

So when the authors refer to "trials" or "trial-to-trial fluctuations" in tuning, what they really have in mind (if I understand correctly) is *slow* changes in tuning that arise over large-ish blocks of time. e.g., In block 1, we present many stimuli that could be used to fit a neuron's tuning curve. Then, in block two, we present the same stimuli again and record a new set of responses, which we can use to re-fit a tuning curve (and observe whether it differed from the tuning curve we fit using block-1 responses).

If this is indeed what the authors have in mind, I believe the paper needs to be substantially re-written to clarify this point. The word "trial" should be removed from the paper, and the authors should make clear that the aim of the paper is to model slow fluctuations in tuning that arise over time. Without such clarification I fear that readers may make the same mistake I did -- thinking that the authors are proposing to do the impossible (namely, model variability of an entire tuning curve on the basis of a single scalar response on each trial!)

2. My second major concern is a lack of quantitative validation of the proposed model. The author are proposing a model that extends / replaces previous models of neural variability (such as Goris 2014 or Charles 2018), in particular by modeling slow-timescale changes in tuning that exhibit low-dimensional structure. However, I did not see a comparison in terms of cross-validated log-likelihood between the proposed model and previous models of spike responses. It seems quite likely to me that the proposed model is indeed better, but it is important to see how much of an improvement it provides (similar to the comparison between Poisson and Modulated Poisson in Goris 2014, Fig 1D).

A related point is that the generative model describes the tuning curve in neighboring blocks of trials as iid, yet I would expect there to be slow fluctuations in the components. Can the authors confirm if this is indeed the case, i.e., that the timecourse of fluctuations in the tuning components exhibits correlations in time?

Other comments:

=====

- I wonder if the authors would consider calling their model a factor models (instead of PCA)? This seems a bit more accurate, since PCA is merely dimensionality reduction and does not provide a generative model (while factor models do). Although I understand the branding appeal of having "PCA" in the title. :/

- Abstract: "scientific essential" -> "scientifically essential"

- pg 6: "whole thing curve"

- pg 6: "we first splitted" -> "split"

- pg 8: "Previous studies have proposed two forms of gain change in V1 [25, 28, 29, 30], i.e. additive and multiplicative."

Another relevant paper to cite here Charles et al ("Dethroning the fano factor". Neural Comp 2018), which also considers additive and multiplicative noise (and is similar to the framework used here in that it uses Gaussian noise transformed by a nonlinearity, followed by Poisson spiking)

- Sec 2.5 (on low-dimensionality of population responses). It would be nice to reflect on whether the authors feel their results conflict with those of Stringer et al ("High-dimensional geometry of population responses in visual cortex" Nature 2019), which found evidence for high-dimensional responses on V1. (Perhaps the difference is that that paper considered natural image stimuli?)

- p 24: "We thus adopt a Monte Carlo approach to solve them." This needs more detail.

1 General remarks

We would like to thank the reviewers for their constructive critiques, which have helped us significantly improve the manuscript. We have revised the manuscript according to their suggestions and questions. Among the various changes we made, major ones include:

1. Adding a detailed discussion of the anesthetized v.s. awake states, as suggested by Reviewer 1.;
2. Providing a further clarification of the concept of tuning variability, and relatedly the terminology of “trial”, which is an issue raised by Reviewer 2;
3. Adding new results showing the quantitative comparison between our method and the Modulated Poisson model as suggested by Reviewer 2. These new results further confirm that our new model indeed accounts for the data better when compared to the prior model.

Changes made during the revision are highlighted in red in the revised manuscript. A point-by-point response to the critiques raised is given below.

2 Point-by-point response

2.1 Reviewer 1 (Remarks to the Author):

Zhu and Wei’s manuscript title ‘unsupervised approach to decomposing neural tuning variability’ seeks a hidden structure of variability within visual neural responses to grating stimuli. They used 7 datasets recorded from anesthetized monkeys. They used a method similar to exponential-family PCA [Collins et al. 2002]. Main differences are independent Gaussian prior on the coefficients, expectation-maximization inference procedure, and cubic spline smoothness regularization. They found that more than 60% of the trial-to-trial tuning curve variability that exceeded the Poisson variability was captured by 1 component. Furthermore, they found that this component was correlated with mean tuning curve. Therefore, they propose a new model of tuning variability which is a Poisson tuning with power-function nonlinearity and 1-dimensional latent fluctuation. However, at a population level, this 1-d fluctuation per neuron were not synchronized, and formed a 1-3 dimensional structure. Two additional analyses, the Fisher information and the representational distance, that showed that the power-law exponents are not consistent with multiplicative model.

The method is straightforward and the story of the paper is presented well. I enjoyed reading the paper. The basic results are impressive and relevant. I’m glad to see a nonparametric approach to the tuning variability analysis.

Response: Thank you for this positive assessment of our work.

Please make the analysis and plotting code available. It would help with reproducibility and specifying some of the implementation details that are omitted in the methods.

Response: Yes. Please be assured that we will make the code available.

Fig 1b is confusing. Shouldn't the mean be a sharp quadratic that reaches extreme negative numbers quickly?

Response: Thank you for this question, and apologize for the confusion. Fig. 1b is a schematic figure illustrating the idea behind the proposed framework in the case of multiplicative gain. Quantitatively, the shape of the main component depends on the particular choice of the tuning curve. If the tuning curve is assumed to be a Gaussian function, then indeed the logarithm of it would be a quadratic form as suggested by the reviewer. In the original version, we neglected this important detail, and it was indeed misleading. We have improved the visualization in Fig 1b to avoid confusions. In the revised version, we set the tuning curve to be a Gaussian function plus a positive baseline (most neurons we analyzed here have a substantial baseline firing rate). The region around the peak follow a quadratic-like decay. Because of the baseline firing, the tail decays much slower than a quadratic function. Similarly, we have also updated Fig 1c. Thank you for catching this and for helping us improving the presentation.

Sec 4.1 eq (5) is missing an overall noise term that corresponds to the σ_0^2 below eq (6)?

Response: Yes, indeed the random noise term was missing. Thank you for pointing this out. We have now added a noise term in the equation together with an explanation of this term.

I'm unfamiliar with [30], and the population FI analysis didn't make much sense to me. I would appreciate if the authors can add why population FI as a function of firing rate is important (other than to show that multiplicative model's predictions do not hold; that point has been made already, right?)

Response: As the reviewer correctly pointed out, one goal of this analysis was indeed to further corroborate the finding that the responses can be not explained by the multiplicative model. We consider that it is useful to support a finding using different approaches, e.g., analysis of tuning curve, analysis of Fisher information (while it carries some redundancy, it is also reassuring). In addition, we believe that this analysis is valuable because understanding how the information varies with the total neural activity is an interesting question by itself. Prior work (e.g., ref[30]) has looked into this question using an approach based on classification. Our results are different from their conclusions in several ways, and our further analysis reveals a weakness of the previous approach. Thus, we consider that this FI analysis carries additional value and is thus useful to be included in the paper. We would be willing to consider taking this analysis out or shorten it, if the reviewer feels strongly about it.

I'm also unfamiliar with [62], and the representational distance, but I only really got the message that the tuning fluctuations and the stimulus contrast prediction which is multiplicative are not consistent. Since the power-law exponents already showed that many of the neurons aren't multiplicative, this isn't surprising, right? I may be missing something.

Response: Thank you for this note. This analysis on representational geometry further supports that the tuning variability of the V1 responses in these experiments is not consistent with a multiplicative gain, as pointed out the reviewer. In addition, it suggests that the representational geometry is also inconsistent with the predictions of a model based on an additive change of tuning curves. Furthermore, we consider that understanding how the latent fluctuation affects the collective population structural of the neural code represents an interesting question on its own, as the geometry of the encoding manifold has multiple implications in understanding the format of the representation and in linking neural responses to the behavior. Our analysis shows that the representation is primarily drifting along an dimension, which is orthogonal to the dimensions that encode the stimulus orientation information. This provides extra information and intuition of how the tuning variability affects the neural response manifold. Thus, we consider this analysis to be a useful addition that compliments the previous analysis. To make this point more explicit, we have edit the text. Now the text reads:

“The geometry of the encoding manifold has multiple implications in understanding the format of the representation and in linking neural responses to the behavior (reviewed in Kriegeskorte & Wei, 2021). How the tuning variability affects the geometrical properties of the encoding manifold represents an interesting yet unresolved question. To investigate this question, we begin by...”

The paper lacks in-depth discussion on anesthetized state. Although the authors indicated that it is a limitation of the current study, description on specifically how it could be limited would be useful.

Response: This is indeed an important point. We have added a more detailed discussion of the anesthetized states v.s. awake states in the Discussion section of the paper. Please see below. Thank you for this suggestion.

“To the extent by which the structure of noise fluctuations under the anesthetized states resembles those of the awake-behaving animals remains an open problem. Earlier work using voltage-sensitive dye to measure large-scale activity fluctuation in V1 under anesthesia found that the structure of the spontaneous fluctuations resembled the stimulus-driven activity, and they interacted with stimulus-evoked activity in an additive fashion [Arieli1996,Kenet2003]. [Goris2014] reported that slow gain fluctuations identified in the anesthetized macaque were also present in the awake state. In addition, results in [Arandia2016] found that the additive and multiplicative change of the tuning curves were also present in a smaller dataset from one macaque monkey. Nonetheless, anesthesia can change the properties of neuron integration in cortical neurons [Suzuki2020], and may trigger a profound change of the cortical dynamics [Alkire2008] and coding [Filipchuk2022]. Detailed in-depth investigations in the future will be important to determine whether the rule of V1 tuning variability that we discovered from anesthetized states may generalize to the awake states. One further limitation of the anesthetized data is that it precludes the analysis of the latent tuning fluctuations with behavior. It would be interesting to see if the fluctuations of internal states similar to what we found here correspond to a change of the behavior[Cowley2020]”.

As the authors noted in the discussion, since spike counts within a (~ 10 sec) block were considered to come from a particular random tuning curve, the fluctuations that are faster (which are typical in anesthesia) are lumped together within the nonlinear model.

Response: Thank you for this remark. This is more of an issue with the current dataset, not a limitation of our method by itself. As we mentioned in the Discussion, there are at least two ways to improve upon the current

results. The first is to analyze datasets with faster stimulus sampling. Second, it should be possible to leverage population structure (e.g., assume the population fluctuation has a low-rank structure) to obtain estimates of the latent variables at an faster timescale. We are developing new methods to pursuing the second direction at the moment, and would like to report the results in an following-up paper.

This is more of a comment than a critique. No action is needed: Would it be possible to re-bin the data at a smaller temporal bin (e.g. 100 or 500 ms) and repeat the analysis? Would it change any of the conclusions?

Response: In this paper, we counted the number of spikes for a 500ms window. Indeed, we checked the performance if we used a shorter 200ms window for each stimulus, and obtained the similar observations.

I understand the smoothness of tuning function would be more difficult to impose in that case. It is probably more appropriate to assume smoothness of the latent variable over time which causes the variability of the tuning curves. As the authors mention, there are many GP based factor models that would be able to tackle exactly this problem. I would also love to see the temporal structure of the fluctuations of the latent. But that's perhaps outside the scope of this paper.

Response: Indeed, this is a very interesting direction that we would like to pursue in the future. In the current algorithm, α_t is assumed to be independent, and is independently calculated for each t . Thus no temporal smoothing is considered. It is possible to extend it to the case with the temporal structure of α_t . One potential way is to use a weighted Poisson fPCA where the weights are constructed by using temporal kernel. Detailed algorithm needs to be investigated. In the revision, we have included this point in the Discussion with the following text:

“Our current method does not explicitly model the temporal structure of the tuning fluctuation, as α_t is assumed to be independent, and is estimated for each t . It should be possible to improve our method by leveraging temporal smoothness prior on the scores, e.g., by weighted Poisson fPCA where the weights are constructed by using temporal kernel, or assuming a Gaussian process prior...”

Minor

- I think 'gain' to be multiplicative, so 'additive gain' is strange to me. I would prefer 'additive modulation', or simply 'additive'.

We agree. Throughout the text, we have change it to "additive modulation", "additive change", or just "additive". Thank you for this suggestion!

- Fig 3, are the 7 "sessions" the 7 "datasets"

Response: Yes. This is now further clarified in the text.

- Fig 3 caption, linear on the mean → scaled?? (rephrase)

- Fig 3 caption, closed → close to
- Fig 5 caption: bock → block
- p12, can not → cannot
- p13, is simple → is a simple

Response: We have fixed these. Thank you for these suggestions.

- p16, what decoder is assumed for the "drift" of representation?

Response: Sorry about the confusion. We would like to clarify that the drift of the presentation is a direct characterization of the representation. No decoder was assumed. One interesting question to consider is what the "drift" means for the downstream area, in which case a particular decoder would need to be assumed to investigate that question. However, that is not the question we seek to address. Here, we merely provide a quantification of the format of the representation.

2.2 Reviewer 2 (Remarks to the Author):

This paper describes a new analysis of response variability of single neurons in visual cortex. Specifically, it uses a latent factor model to characterize slow changes in a neuron's tuning curve over time in terms of a small number of tuning-curve-like components modulated by Gaussian noise (and then passed through a nonlinearity). The resulting model can identify variable firing patterns over time as resulting from shifts in a tuning curve, tuning curve width changes, and additive or multiplicative response noise. The authors also provide novel analyses of the coding implications of this form of variability using Fisher information. I think the proposed framework interesting and important, likely to be of interest to a large segment of the wider field.

Response: Thank you for your positive assessment of our work.

However, there are two major shortcomings that should be addressed before the paper is suitable for publication.

1. The first problem is conceptual. The phrase "trial-to-trial tuning variability" (which appears in the abstract) is non-sensical (at least as I read it) given the definition of tuning curve, which is that it is the average or expected response to a stimulus. By definition, the tuning curve cannot fluctuate from trial to trial. The tuning curve IS the average response across trials, and the trial-to-trial fluctuations are (commonly known as) noise (NOT tuning).

At first I thought this was merely a terminology problem, but I grew more alarmed when I got to equation 1, which describes a factor model for single-neuron responses. This appears even more deeply nonsensical, since a single neuron responds only to the single stimulus presented on each trial. We can't model a vector of responses from a single neuron to multiple stimuli on a single trial! This makes no sense. A neuron can only respond to one stimulus at a time, so this seems tantamount to attempting to model noise correlations between neurons that were not recorded simultaneously.

It was only when I finally got to the Methods section of the paper that I realized that authors have something very different in mind when they refer to "a trial": pg 25: "Because the experiments had a block-randomized design, for each block we obtained a response vector corresponding the responses for all the stimulus orientations sampled in the experiments. Repeating this for every block, we constructed a spike count matrix for each neuron."

So when the authors refer to "trials" or "trial-to-trial fluctuations" in tuning, what they really have in mind (if I understand correctly) is *slow* changes in tuning that arise over large-ish blocks of time. e.g., In block 1, we present many stimuli that could be used to fit a neuron's tuning curve. Then, in block two, we present the same stimuli again and record a new set of responses, which we can use to re-fit a tuning curve (and observe whether it differed from the tuning curve we fit using block-1 responses).

If this is indeed what the authors have in mind, I believe the paper needs to be substantially re-written to clarify this point. The word "trial" should be removed from the paper, and the authors should make clear that the aim of the paper is to model slow fluctuations in tuning that arise over time. Without such clarification I fear that readers may make the same mistake I did – thinking that the authors are proposing to do the impossible (namely, model variability of an entire tuning curve on the basis of a single scalar response on each trial!)

Response: Thank you for this thoughtful comment! Yes, indeed by the "trial", we were referring to the blocks for the V1 experiments.

There are two perspectives to think about "tuning variability". First, from a theoretical point of view, at any moment in time, we could "freeze" and probe the system with any set of stimuli and obtain a tuning curve which provides a quantitative description of how the system (i.e., individual neurons) responds to any external stimulus. We can then consider the question of how the tuning properties vary from moment-to-moment. Second and practically, as the reviewer correctly pointed out, we need multiple measurements to get a reasonable estimate of the tuning curve, and these measurements can only be obtained sequentially (unlike the theoretical perspective mentioned above). In this case, one need to make the assumption that the tuning curve is fixed within a time window during which period we could gather sufficient number of measurements. The size of the time window in a given experiment would depend on the particular design. For the V1 experiments we analyzed, each block of trials would be a reasonable choice (10 seconds) for estimating the tuning curve. Note that the relatively long time scale (~ 10 seconds) is not a limitation the method itself. If the experiment has a much faster sampling rate, e.g., 100ms per stimulus, we could apply the same method to estimate the tuning variability at the timescale of ~ 1 second.

We agree with the reviewer that "trial-to-trial tuning variability" is confusing as many would equate a "trial" with a single stimulus. We have thus changed it to "moment-to-moment tuning variability" throughout the text. We believe this is a more general and also more accurate terminology. Moment could refer to different timescales depending on the particular experiments. We have made these points explicit in the revised text:

In Section 2.3. "*We treated each block of stimuli as one trial, assuming that the tuning curve is stable within each block. Thus the tuning fluctuations studied here is at the timescale of ~ 10 seconds.*"

In the Discussion, we added more discussions to address this point: "*...our method, when applying to V1 data, only deals with slow fluctuations (~ 10 seconds) [Goris2014], because of the assumption that the latent is the same within every block (or moment). Thus, this inferred moment-to-moment tuning fluctuation is at the time scale of ~ 10 seconds. Tuning variability at even faster time scale would be averaged out. Thus our estimate of the tuning*

fluctuation is likely an under-estimate of the true fluctuations. It should be possible to refine these estimates and study tuning variability at even faster scale. Two approaches may be promising: i) by using faster stimulus sampling in experiments—with a stimulus sampling of 100ms per stimulus, it is possible to apply the same approach to study tuning variability at a timescale of ~ 1 second...

We believe this helps further clarify the contribution of the work. We would like to thank the reviewer for this suggestion.

2. My second major concern is a lack of quantitative validation of the proposed model. The author are proposing a model that extends / replaces previous models of neural variability (such as Goris 2014 or Charles 2018), in particular by modeling slow-timescale changes in tuning that exhibit low-dimensional structure. However, I did not see a comparison in terms of cross-validated log-likelihood between the proposed model and previous models of spike responses. It seems quite likely to me that the proposed model is indeed better, but it is important to see how much of an improvement it provides (similar to the comparison between Poisson and Modulated Poisson in Goris 2014, Fig 1D).

Response: This is an excellent suggestion. Following this suggestion, we have performed quantitative comparisons with Goris et al 2014. In the revision, we have supplied the quantitative validation. We performed a K -fold cross-validation, where K is the length of stimuli. We also compared the conditional probability given the predicted $\hat{\lambda}_i(s)$ of the hold-out data. By these measures, we compared our method with Modulated Poisson in Goris 2014. We now mentioned this in the text, and include detailed results in SI Fig.2. We feel that these additional results further strengthen the paper. Thank you for this suggestion.

In addition, it is worth mentioning that our method enables recovery of the score for each block. This is not possible in the Modulated Poisson proposed in Goris 2014.

A related point is that the generative model describes the tuning curve in neighboring blocks of trials as iid, yet I would expect there to be slow fluctuations in the components. Can the authors confirm if this is indeed the case, i.e., that the timecourse of fluctuations in the tuning components exhibits correlations in time?

Response: Thank you for this suggestion. We have performed this analysis, and included the auto-correlation function of scores as a new figure in the SI (SI Fig. 7). The result shows that auto-correlation varies smoothly, beyond the time scale of 10s. The actual temporal correlation may be even stronger, as the noise during the estimation of the score will reduce the magnitude of the estimated autocorrelation.

Other comments: =====

- I wonder if the authors would consider calling their model a factor models (instead of PCA)? This seems a bit more accurate, since PCA is merely dimensionality reduction and does not provide a generative model (while factor models do). Although I understand the branding appeal of having "PCA" in the title. :/

Response: This is indeed an interesting point to visit. Although perhaps not the conventional way to think about PCA, it's previously demonstrated that PCA could be interpreted as a generative model with vanishing homeostatic Gaussian noise, e.g., see the following reference: M.E. Tipping and C.M. Bishop (1999), Probabilistic principal component analysis, J. R. Statist. Soc. B . Our model can be viewed as a generalization of PCA to Poisson noise and functional data. In PCA, the components are orthogonal linear combinations that maximize the total variance, and the loadings (α_t in our notations) present how much each variable contributes to a particular principal component. This matches well with our motivation, as we want to decompose neural tuning variability along different orthogonal components. In contrast, factor analysis (FA) analyzes only the reliable common variance of data by maximizing the shared portion of the variance. This is conceptually different from the goal of our proposed method. Thus, while we appreciate the reviewer's suggestion, after thinking more about it we still feel that it would be more appropriate to view our method as an extension of PCA.

- **Abstract: "scientific essential" → "scientifically essential"; - pg 6: "whole thing curve"**

Response: We have corrected them. Thank you very much.

- **pg 6: "we first splitted" → "split"**

Response: Fixed.

- **pg 8: "Previous studies have proposed two forms of gain change in V1 [25, 28, 29, 30], i.e. additive and multiplicative." Another relevant paper to cite here Charles et al ("Dethroning the fano factor". Neural Comp 2018), which also considers additive and multiplicative noise (and is similar to the framework used here in that it uses Gaussian noise transformed by a nonlinearity, followed by Poisson spiking)**

Response: Thank you for this suggestion. This is indeed a relevant paper to cite. We have added this paper, and referred to it several times when appropriate.

- **Sec 2.5 (on low-dimensionality of population responses). It would be nice to reflect on whether the authors feel their results conflict with those of Stringer et al ("High-dimensional geometry of population responses in visual cortex" Nature 2019), which found evidence for high-dimensional responses on V1. (Perhaps the difference is that that paper considered natural image stimuli?)**

Response: This is an interesting question. The Stringer paper analyzed the tail of the PCs, while we focus on the first few PCs. Notice that our datasets only have ~ 100 cells. Due to the limitation of the data, we are not in a position to really address this question. We will add a detailed discussion on this point:

"Our results suggest that the tuning fluctuation exhibits low-rank structure, both at the levels of individual neurons and neural populations. The latter is generally consistent with, and generalizes from, results in previous studies that assumed coherent gain fluctuations among simultaneously recorded neurons [Lin2015, Scholvinck2015, Rabinowitz2015]. A recent study [Stringer2019] found that the variance explained by the PCs of large-scale neural populations scaled as a power-law. Our results are different from theirs: (i) our results concern the dimensionality of the tuning variability, not the dimensionality of stimulus tuning; (ii) we primarily focus on the amount of variance explained by the first few PCs, not the properties of the tail of the spectrum as done in [Stringer2019]. With a few dozens of simultaneously recorded neurons, we can not accurately estimate the scaling relationship between the variance (of

the tuning variability) explained of the neural population and the number of PCs– an interesting question that could be addressed in the future with larger datasets. ”

- p 24: ”We thus adopt a Monte Carlo approach to solve them.” This needs more detail.

Response: In the revision, we have added more details on the Monte Carlo approach we used in the Method Section.

We now write:

*“We thus adopt a Monte Carlo approach to calculate them. For each t , we generate a set of samples, $\log(\vec{\mu}_t)^{*1}, \dots, \log(\vec{\mu}_t)^{*M}$, where $M = 10,000$ is the number of Monte Carlo runs, according to the distribution of $\log(\vec{\mu}_t)$ given by the current parameters. Then the unbiased estimates are obtained from the samples we simulated. Together, **Step 1** gives an estimator of hidden means $\vec{\mu}_t$ in the form of $E[\log(\vec{\mu}_t)|\vec{n}_t]$.”*

REVIEWERS' COMMENTS

Reviewer #1 (Remarks to the Author):

Thank you for updating the manuscript and answering the points raised.

I have an additional request for the analysis: since this new statistical method is working well for the given dataset, it would be important to know how well it scales with the amount of data provided. As a function of stimulus presentations and/or blocks, how does Pf-PCA behave in terms of the main finding, i.e., the (1) deviation in the mean component, (2) proportion of variance explained by the first fPC, and (3) the percentage of first fPC resembling the mean component (corresponding to Fig 3)? Such result would encourage replication studies to know the appropriate amount of data needed for the conclusions.

Also, please answer the following clarifying questions:

- Citation needed: "a recent study proposed that gain variability in V1 serves to represent the stimulus uncertainty via sampling, a computation would generally require the gain variability to be high dimensional."

- Rank-1 model: details on how the statistics were matched with the real data is missing (methods section, Fig 5 and S9, S10, S11). To make the conclusion that "The empirically observed latent fluctuations cannot be explained by a rank-1 multiplicative or additive modulation model.", it is important to match signal-to-noise ratio (assuming the amount of data and firing rates are matched, if not please specify), since with less data or more "noise", the statistical difference would disappear. (Also, it is curious that from Fig S9, the 2nd dataset (D2?) seem quite low-rank! Just an observation.)

- How were the population FI plots as a function of spike counts made? If each dot on Fig 6c does not correspond to a single stimulus presentation, how are they aggregated? Are the trends in Fig 6c and 6d dominated by a few neurons in each session or were there patterns across the neurons? According to the methods they were simply summed over neurons and orientation, and the origin of the FI-modulation index is not clear to me.

Reviewer #2 (Remarks to the Author):

The authors have done a nice job responding to my comments, and I thank them for the careful and detailed replies. I believe the manuscript is greatly improved and is now ready for publication.

For the record I still believe that it would have been more accurate to refer this as a Poisson factor model or factor-analytic model instead of a PCA — the probabilistic PCA model of Tipping & Bishop is simply factor analysis with the per-neuron diagonal noise covariance matrix set to a multiple of the identity matrix... so it has more in common with factor modeling than PCA, in my humble opinion. But I defer to the authors and feel they should be free to call it what they wish to. (Note the model also bears a lot of similarity to the vLGP model from Zhao & Park 2017 [95]).

Congrats on an excellent paper and I look forward to seeing it in print!

1 General remarks

We would like to thank the reviewers for positive assessment on our previous revision. In this Final revision, we have further revised the manuscript according to their suggestions. A point-by-point response is given below. Changes made during the revision are highlighted in red in the main text of the manuscript. Changes made to the Supplementary Information (SI) is described in the point-by-point response.

2 Point-by-point response

2.1 Reviewer #1 (Remarks to the Author):

I have an additional request for the analysis: since this new statistical method is working well for the given dataset, it would be important to know how well it scales with the amount of data provided. As a function of stimulus presentations and/or blocks, how does Pf-PCA behave in terms of the main finding, i.e., the (1) deviation in the mean component, (2) proportion of variance explained by the first fPC, and (3) the percentage of first fPC resembling the mean component (corresponding to Fig 3)? Such result would encourage replication studies to know the appropriate amount of data needed for the conclusions.

Response: Thank you for this suggestion. We have added an analysis to examine the impact of the sample size. For each dataset, we ran Pf-PCA on subsets of the data with 25, 50, and 100 blocks respectively. Based on the results (see the new Fig. S7 and Section SI.5), we observed that the number of blocks used only had a small impact on the proportion of variance explained by the first fPC and the significance of regression testing. However, reducing the sample size has an more substantial impact on the “fraction” index regressing the first fPC against the recovered mean component. Thus, while our results are reasonably robust to different sample sizes, having larger sample sizes remains advantageous.

- Citation needed: ”a recent study proposed that gain variability in V1 serves to represent the stimulus uncertainty via sampling, a computation would generally require the gain variability to be high dimensional.”

Response: Thank you for pointing this out. We have added the appropriate citation.

- Rank-1 model: details on how the statistics were matched with the real data is missing (methods section, Fig 5 and S9, S10, S11). To make the conclusion that ”The empirically observed latent fluctuations cannot be explained by a rank-1 multiplicative or additive modulation model.”, it is important to match signal-to-noise ratio (assuming the amount of data and firing rates are matched, if not please specify), since with less data or more “noise”, the statistical difference would disappear. (Also, it is curious that from Fig S9, the 2nd dataset (D2?) seem quite low-rank! Just an observation.)

Response: Thank you for this question. To ensure comparability of results, we set the variance of multiplicative gain fluctuations in such way that the fluctuations of each neuron match the values in real data. We have added a more detailed description of this procedure in Section 4.6. .

- How were the population FI plots as a function of spike counts made? If each dot on Fig 6c does not correspond to a single stimulus presentation, how are they aggregated? Are the trends in Fig 6c and 6d dominated by a few neurons in each session or were there patterns across the neurons? According to the methods they were simply summed over neurons and orientation, and the origin of the FI-modulation index is not clear to me.

Response: Thank you for these questions.

Figure 6c was obtained by, within each block for each session, computing (i) the total FI by summing over all orientations and all neurons; and (ii) the total number of spikes. Then the two quantities were visualized using a scatter plot. We have further clarified this procedure in the Methods section 4.7.

There was a figure (Fig. S14 in the current version) reporting the relationship of FI and spike counts for individual neurons in one example dataset. Fig. 6b reports the histogram of the slope of the FI-spike count curve for all neurons analyzed. In addition, we have now added a new figure (Fig. S13) that reports the histograms of the slope of the FI-spike count curve for individual datasets. Given these results, it appears that the effects reported in Fig. 6c and 6d are distributed in the neural population.

2.2 Reviewer #2 (Remarks to the Author):

The authors have done a nice job responding to my comments, and I thank them for the careful and detailed replies. I believe the manuscript is greatly improved and is now ready for publication.

For the record I still believe that it would have been more accurate to refer this as a Poisson factor model or factor-analytic model instead of a PCA — the probabilistic PCA model of Tipping & Bishop is simply factor analysis with the per-neuron diagonal noise covariance matrix set to a multiple of the identity matrix... so it has more in common with factor modeling than PCA, in my humble opinion. But I defer to the authors and feel they should be free to call it what they wish to. (Note the model also bears a lot of similarity to the vLGP model from Zhao & Park 2017 [95]).

Congrats on an excellent paper and I look forward to seeing it in print!

Response: Thank you for your positive assessment of our last revision, and your suggestions that have helped us greatly improved the paper.